# Predatory protists reduce bacteria wilt disease incidence in tomato plants

Sai Guo ©[1,2], Zixuan Jiao ©[1], Zhiguang Yan ©[1], Xinyue Yan ©[1], Xuhui Deng[1,2], Wu Xiong ©[1,2], Chengyuan Tao ©[1,2], Hongjun Liu[1,2], Rong Li ©[1,2] ✉, Qirong Shen ©[1], George A. Kowalchuk ©[3] & Stefan Geisen[4,5]

Soil organisms are affected by the presence of predatory protists. However, it remains poorly understood how predatory protists can affect plant disease incidence and how fertilization regimes can affect these interactions. Here, we characterise the rhizosphere bacteria, fungi and protists over eleven growing seasons of tomato planting under three fertilization regimes, i.e conventional, organic and bioorganic, and with different bacterial wilt disease incidence levels. We find that predatory protists are negatively associated with disease incidence, especially two ciliophoran *Colpoda* OTUs, and that bioorganic fertilization enhances the abundance of predatory protists. In glasshouse experiments we find that the predatory protist *Colpoda* influences disease incidence by directly consuming pathogens and indirectly increasing the presence of pathogen-suppressive microorganisms in the soil. Together, we demonstrate that predatory protists reduce bacterial wilt disease incidence in tomato plants via direct and indirect reductions of pathogens. Our study provides insights on the role that predatory protists play in plant disease, which could be used to design more sustainable agricultural practices.

Crop losses caused by soil-borne pathogens are becoming an ever-increasing threat to sustainable agricultural production[1,2]. Pathogens (e.g., *Ralstonia solanacearum* and *Fusarium oxysporum*) severely impact plant health when colonizing the plant rhizosphere[3]. The plant rhizosphere provides numerous ecological niches for the growth and proliferation of a variety of microorganisms[4]. Rhizosphere microorganisms, especially bacteria and fungi, form a tight network to improve plant health by defending against pathogens through competition for space and resources[5,6]. Negative impacts on rhizosphere microorganisms, such as through intense management, therefore often are reported to increase plant disease incidences and reduce plant health[7,8].

Healthy plants are essential for ensuring crop productivity and food security[9]. Conventional agricultural management is linked to high

management efforts to inhibit pests, but the intensive application of synthetic agrochemicals (e.g., chemical fertilizers and synthetic pesticides) in conventional agriculture has adverse effects on soil quality and environmental sustainability[10,11]. Organic farming, such as organic fertilizer application, can improve plant performance (e.g., growth and health) and minimize negative impacts of synthetic chemicals by inducing beneficial bacteria and fungi as well as their ecological interactions in soils and plant rhizospheres[8,12,13]. Through the promotion of plant-beneficial bacteria and fungi, organic farming is recognized as one of the most sustainable alternatives to conventional agriculture[11,14].

Bacteria and fungi are also top-down controlled by predators, particularly predatory protists that are the dominant soil protists[15]. In

[1]Jiangsu Provincial Key Lab of Solid Organic Waste Utilization, Jiangsu Collaborative Innovation Center of Solid Organic Wastes, Educational Ministry Engineering Center of Resource-Saving fertilizers, Nanjing Agricultural University, Nanjing 210095 Jiangsu, PR China. [2]The Sanya Institute of the Nanjing Agricultural University, Sanya, Hainan Province, PR China. [3]Ecology and Biodiversity Group, Department of Biology, Institute of Environmental Biology, Utrecht University, Padualaan 8, 3584 CH Utrecht, The Netherlands. [4]Laboratory of Nematology, Wageningen University, 6700 AA Wageningen, The Netherlands. [5]Netherlands Department of Terrestrial Ecology, Netherlands Institute for Ecology, (NIOO-KNAW), 6708 PB Wageningen, The Netherlands. ✉e-mail: lirong@njau.edu.cn

turn, bacteria and fungi affect predatory protists. For example, anti-predatory compounds released by bacteria and fungi commonly inhibit protists[16,17]. Also species-specific interactions between protist predators and microbial prey are common, as predators specifically select for their preferred prey microorganisms[15,16,18]. Predatory protists are also directly affected by changes in the physicochemical surrounding (e.g., soil moisture availability and pH)[19,20]. Therefore, predatory protist communities are sensitive to fertilizer application, potentially even more than their microbial prey (e.g., bacteria and fungi)[21,22]. Predatory protists stimulate nutrient turnover (e.g., nitrogen) resulting in increased plant-availability[16]. Predatory protists also shift microbiome structure and functionality, often leading to enhanced soil fertility and plant yield[21–23]. While an increasing number of studies uncover the importance of protists in the complexity of disease suppression and therefore as contributors to plant health[15,16], the mode of action how these predatory protists increase plant health remains unknown.

To investigate microbial mechanisms underlying fertilization-induced increases in plant health, we determined rhizosphere bacteria, fungi and predatory protists in long-term tomato fields under conventional, organic, and bioorganic fertilization regimes with different bacterial wilt disease incidences. Subsequently, we performed greenhouse experiments to validate observed direct (induced by predatory protists) and indirect (induced by microbial prey of predatory protists) effects of key predatory protistan taxa on the suppression of the major tomato bacterial wilt pathogen *Ralstonia solanacearum* (*R. solanacearum*) in the field experiment and to explore the potential bacterial wilt disease-suppressive mechanisms. We hypothesized that (1) protistan communities are more strongly affected by bioorganic fertilization than bacterial and fungal communities, (2) predatory protists explain the decrease in disease incidences and pathogen densities associated with bioorganic fertilization better than other microbial groups, (3) predatory protists directly suppress *R. solanacearum* through consumption and (4) indirectly improve plant health by promoting pathogen-suppressive microorganisms.

## Results

### Bacterial, fungal, and protistan communities and their contribution to bacterial wilt disease incidence

The long-term field experiment was performed with different fertilization treatments (conventional, organic, and bioorganic fertilization) that resulted in different disease incidence. Fertilization regimes had a greater impact on the diversity (one-way ANOVA: $F = 7.28$, $p = 0.025$) and community composition (PERMANOVA: $R^2 = 0.76$, $p = 0.025$) of

protists compared to bacteria and fungi (bacterial diversity: one-way ANOVA: $F = 0.51$, $p = 0.625$, fungal diversity: one-way ANOVA: $F = 0.99$, $p = 0.426$; bacterial community composition: PERMANOVA: $R^2 = 0.37$, $p = 0.074$, fungal community composition: PERMANOVA: $R^2 = 0.26$, $p = 0.425$) (Supplementary Fig. 1). The explanatory power of protistan diversity and community composition (linear model analysis: explaining 86.36% of the observed variation) for disease incidence was higher than that of bacterial and fungal diversities and community compositions (linear model analysis: bacterial diversity and community composition: explaining 10.26% of the observed variation; fungal diversity and community composition: explaining 2.61% of the observed variation, Fig. 1A). Among the selected microbial indices, protistan community composition best explained disease incidence (linear model analysis: explaining 45.36% of the observed variation, $F = 299.32$, $p = 0.004$; Fig. 1A).

Among protistan functional groups, only the relative abundances of predatory protists were negatively correlated with disease incidence (Spearman's correlation analysis: correlation coefficient = −0.82, $p = 0.007$, Fig. 1B and Supplementary Table 1). In addition, predatory protists were correlated with the bacterial community (Mantel correlation analysis: $r = 0.52$, $p = 0.018$; Fig. 1C). The relative abundances of predatory protists were highest in BF (bioorganic fertilization) (enhanced by 12.7% in BF and by 8.3% in OF (organic fertilization) compared to CF (conventional fertilization)) (one-way ANOVA with Tukey's HSD test: $F = 5.73$, $p = 0.041$, Fig. S2). As predatory protists were linked to disease incidence and bacteria and were affected by fertilization regimes, we focused subsequent analyses on predatory protists and their links with bacteria.

### Predatory protistan and bacterial taxonomic compositions and links with bacterial wilt disease suppression

The relative abundances of seven predatory protistan OTUs negatively correlated with disease incidence, including three ciliophoran, two cercozoan, one pseudofungal and one conosan taxa (Spearman's correlation analysis: P_OTU67: correlation coefficient = −0.90, $p = 0.002$; P_OTU37: correlation coefficient = −0.72, $p = 0.037$; P_OTU30: correlation coefficient = −0.63, $p = 0.046$; P_OTU39: correlation coefficient = −0.71, $p = 0.032$; P_OTU79: correlation coefficient = −0.75, $p = 0.026$; P_OTU68: correlation coefficient = −0.80, $p = 0.013$; P_OTU59: correlation coefficient = −0.86, $p = 0.002$; Fig. 2A). Of these predatory protistan OTUs, the explanatory power of ciliophoran OTUs (linear model analysis: explaining 41.64% of the observed variation) for the density of *R. solanacearum* was higher than that of cercozoan, pseudofungal and conosan OTUs (linear model analysis:

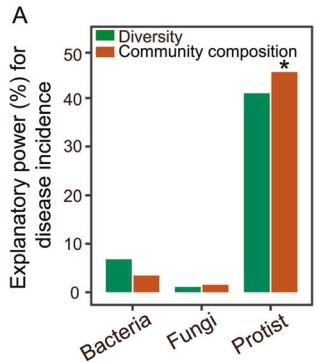
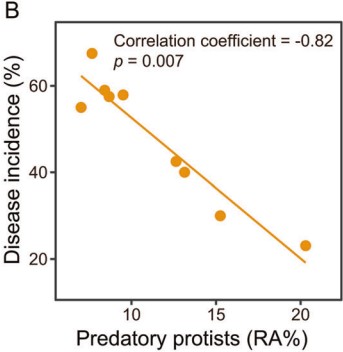
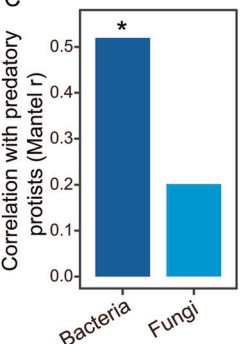

**Fig. 1 | Bacterial, fungal, and protistan communities and their contribution to bacterial wilt disease incidence.** The explanatory power of bacterial, fungal and protistan diversities and community compositions for disease incidence (**A**). Correlations between the relative abundances of predatory protists and disease incidence (**B**). Correlations between bacterial and fungal communities and the relative abundances of predatory protists (**C**). In (**A**), statistical significance was calculated by multiple regression using linear models. Asterisk denotes statistically significant ($p < 0.05$). In (**B**), two-sided Spearman's correlation was performed to explore the relationship between disease incidence and the relative abundance of predatory protists. Solid lines denote statistically significant ($p < 0.05$). In (**C**), statistical significance was calculated by Mantel test. Asterisk denotes statistically significant ($p < 0.05$). Source data are provided as a Source Data file.

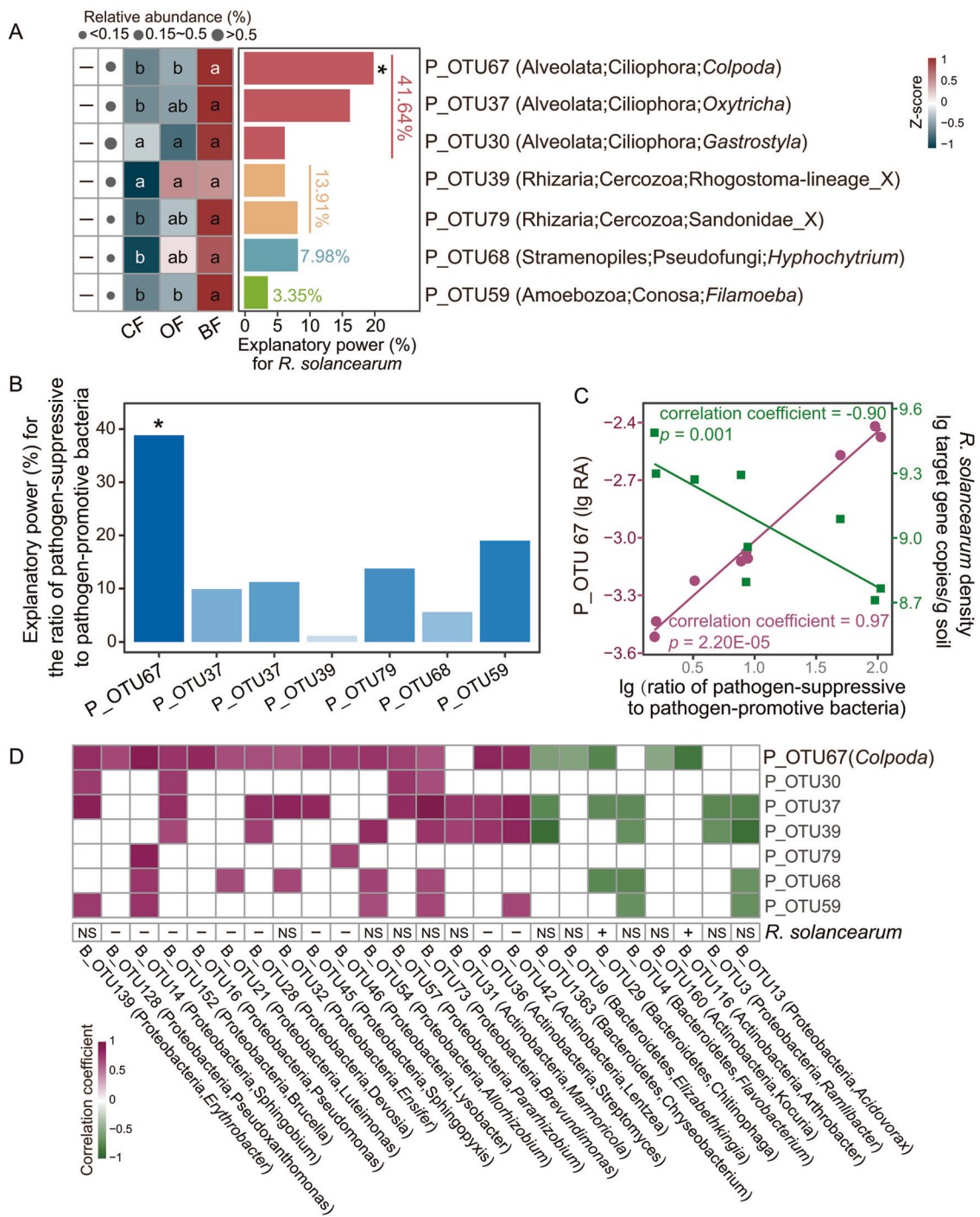

cercozoa: explaining 13.91% of the observed variation; pseudofungi: explaining 7.98% of the observed variation; conosa: explaining 3.35% of the observed variation, Fig. 2A). In particular, the ciliophoran *Colpoda* P_OTU67 was the strongest predictor with respect to explaining the density of *R. solancearum* (linear model analysis: explaining 19.67% of the observed variation, $F = 15.53$, $p = 0.003$, Fig. 2A). The relative abundance of *Colpoda* P_OTU67 was negatively correlated with the density of *R. solanacearum* (Spearman's correlation analysis: correlation coefficient $= -0.93$, $p = 2.36E-04$, Supplementary Table 2). The relative abundance of *Colpoda* P_OTU67 was highest in BF (enhanced by 676.82% in BF and by 86.91% in OF compared to CF) (one-way ANOVA with Tukey's HSD test: $F = 63.94$, $p = 9.00E-05$, Fig. 2A). In addition, *Colpoda* P_OTU67 had the greatest explanatory power for the ratio of pathogen-suppressive bacteria to pathogen-promotive

bacteria (linear model analysis: explaining 38.84 % of the observed variation, $F = 183.38$, $p = 0.047$, Fig. 2B). The relative abundance of *Colpoda* P_OTU67 positively correlated with the ratio of pathogen-suppressive bacteria to pathogen-promotive bacteria (Spearman's correlation analysis: correlation coefficient $= 0.97$, $p = 2.20E-05$) and this ratio negatively correlated with the density of *R. solanacearum* (Spearman's correlation analysis: correlation coefficient $= -0.90$, $p = 0.001$, Fig. 2C). The relative abundance of *Colpoda* P_OTU67 positively correlated with the relative abundances of *Pseudoxanthomonas* (B_OTU128), *Sphingobium* (B_OTU14), *Brucella* (B_OTU152), *Pseudomonas* (B_OTU16), *Luteimonas* (B_OTU21), *Devosia* (B_OTU28), *Sphingopyxis* (B_OTU45), *Lysobacter* (B_OTU46), *Streptomyces* (B_OTU36) and *Lentzea* (B_OTU42) (Spearman's correlation analysis: B_OTU128: correlation coefficient $= 0.70$, $p = 0.037$; B_OTU14: correlation

**Fig. 2 | Predatory protistan and bacterial taxonomic compositions and links with bacterial wilt disease suppression.** Heatmap illustrating the relative abundances of predatory protistan OTUs (linked with disease incidence) in all treatments and the explanatory power of these predatory protistan OTUs for the density of *R. solanacearum* (**A**). The explanatory power of predatory protistan OTUs (linked with disease incidence) for the ratio of pathogen-suppressive to pathogen-promotive bacteria (**B**). Correlations between the relative abundance of *Colpoda* P_OTU67, the ratio of pathogen-suppressive to pathogen-promotive bacteria and the density of *R. solanacearum* (**C**). Correlations between the relative abundances of predatory protistan OTUs (linked with disease incidence) and bacterial OTUs (**B**) and the density of *R. solanacearum* (**D**). In (**A**), P = predatory protist. Statistical significance was calculated by multiple regression using linear models. Asterisk denotes statistically significant ($p < 0.05$). The color key relates heatmap colors to the standard score (z-score). Two-sided Spearman's correlation was performed to explore the relationship between the relative abundances of predatory protistan OTUs and disease incidence. Minus signs denote significant negative Spearman's correlations between the relative abundances of predatory protistan OTUs and disease incidence ($p < 0.05$). Circles are proportional to the average relative abundances of predatory protistan OTUs across all samples. ANOVA with two-sided Tukey's multiple comparison was used for the statistical analysis. Letters: significant differences between treatments ($p < 0.05$). CF conventional fertilization, OF organic fertilization, BF bioorganic fertilization. In (**B**), P = predatory protist. Statistical significance was calculated by multiple regression using linear models. Asterisk denotes statistically significant ($p < 0.05$). In (**C**), P = predatory protist. RA = relative abundance. Two-sided Spearman's correlation was performed to explore the relationship between the relative abundance of *Colpoda* P_OTU67, the ratio of pathogen-suppressive to pathogen-promotive bacteria and the density of *R. solanacearum*. Solid lines denote statistically significant ($p < 0.05$). In (**D**), P = predatory protist, B = bacteria. Two-sided Spearman's correlation was performed to explore the relationship between the relative abundances of bacterial OTUs and the density of *R. solanacearum*. Plus signs denote significant positive Spearman's correlations between the relative abundances of bacterial OTUs and the density of *R. solanacearum* ($p < 0.05$). Minus signs indicate significant negative Spearman's correlations between the relative abundances of bacterial OTUs and the density of *R. solanacearum* ($p < 0.05$). NS indicates no significant Spearman's correlations between the relative abundances of bacterial OTUs and the density of *R. solanacearum* ($p > 0.05$). The heatmap shows significant correlations (calculated by two-sided Spearman's correlation, $p < 0.05$) between the relative abundances of predatory protistan OTUs and the relative abundances of bacterial OTUs. Source data are provided as a Source Data file.

coefficient = 0.92, $p = 0.001$; B_OTU152: correlation coefficient = 0.78, $p = 0.014$; B_OTU16: correlation coefficient = 0.83, $p = 0.008$; B_OTU21: correlation coefficient = 0.67, $p = 0.048$; B_OTU28: correlation coefficient = 0.67, $p = 0.049$; B_OTU45: correlation coefficient = 0.77, $p = 0.021$; B_OTU46: correlation coefficient = 0.74, $p = 0.023$; B_OTU36: correlation coefficient = 0.85, $p = 0.004$; B_OTU42: correlation coefficient = 0.83, $p = 0.008$; Fig. 2D). In turn, the relative abundances of these bacterial OTUs negatively correlated with the density of *R. solanacearum* (Spearman's correlation analysis: B_OTU128: correlation coefficient = $-0.67$, $p = 0.049$; B_OTU14: correlation coefficient = $-0.78$, $p = 0.017$; B_OTU152: correlation coefficient = $-0.67$, $p = 0.048$; B_OTU16: correlation coefficient = $-0.87$, $p = 0.005$; B_OTU21: correlation coefficient = $-0.67$, $p = 0.049$; B_OTU28: correlation coefficient = $-0.70$, $p = 0.043$; B_OTU45: correlation coefficient = $-0.83$, $p = 0.008$; B_OTU46: correlation coefficient = $-0.80$, $p = 0.014$; B_OTU36: correlation coefficient = $-0.68$, $p = 0.047$; B_OTU42: correlation coefficient = $-0.86$, $p = 0.005$; Fig. 2D). In addition, the relative abundance of *Colpoda* P_OTU67 negatively correlated with the relative abundances of *Chitinophaga* (B_OTU29) and *Arthrobacter* (B_OTU116) (Spearman's correlation analysis: B_OTU29: correlation coefficient = $-0.78$, $p = 0.017$; B_OTU116: correlation coefficient = $-0.87$, $p = 0.005$; Fig. 2D), which all positively correlated with the density of *R. solanacearum* (Spearman's correlation analysis: B_OTU29: correlation coefficient = 0.72, $p = 0.037$; B_OTU116: correlation coefficient = 0.72, $p = 0.037$; Fig. 2D).

## Direct effects of the key predatory protists on suppressing pathogens

Of all isolated *Colpoda* strains, *Colpoda* 2, most closely related to *Colpoda inflata* (Supplementary Table 3), had the most similar sequence identity (99.71%) with the key *Colpoda* P_OTU67 of the above field experiment (Supplementary Table 4). Co-inoculation of different *Colpoda* strains and *R. solanacearum* combinations decreased the density of *R. solanacearum* (*Colpoda* 2 + *R. solanacearum*: decrease of 51.08%, *Colpoda* 1 + *R. solanacearum*: decrease of 46.46%) compared to *R. solanacearum* inoculation in sterilized soils (one-way ANOVA with Tukey's HSD test: $F = 129.43$, $p = 3.46\text{E-}10$; Fig. 3A). The relative change of the density of *R. solanacearum* in *Colpoda* 2 + *R. solanacearum* was higher than that in *Colpoda* 1 + *R. solanacearum* in sterilized soils (Student's *t* test: $T = -2.28$, $p = 0.046$; Fig. 3B). As *Colpoda* 2 had the most similar sequence identity with the key *Colpoda* P_OTU67 and had the strongest directly suppressive effect on *R. solanacearum*, we further performed a greenhouse experiment with different concentrations of *Colpoda* 2 in nonsterilized soils to detect the indirect effect of the key predatory protist on promoting plant health by altering microbial communities.

## Indirect effects of the key predatory protists on suppressing pathogens

Compared with the control treatment, the different concentrations of *Colpoda* 2 decreased disease incidence (*Colpoda* 2 ($10^2$): decrease of 34.21%, *Colpoda* 2 ($10^4$): decrease of 55.26%, one-way ANOVA with Tukey's HSD test: $F = 31.20$, $p = 5.00\text{E-}06$; Fig. 4A) and the density of *R. solanacearum* (*Colpoda* 2 ($10^2$): decrease of 82.04%, *Colpoda* 2 ($10^4$): decrease of 96.07%, one-way ANOVA with Tukey's HSD test: $F = 220.96$, $p = 7.45\text{E-}12$; Fig. 4B) in nonsterilized soils. The density of *R. solanacearum* positively correlated with disease incidence (Spearman's correlation analysis: correlation coefficient = 0.76, $p = 2.68\text{E-}04$; Supplementary Fig. 3). *Colpoda* 2 inoculation altered bacterial diversity (one-way ANOVA: $F = 7.81$, $p = 0.005$; Supplementary Figs. 4A and 4C) and community composition (PERMANOVA: $R^2 = 0.75$, $p = 0.001$; Supplementary Fig. 4B), but not of fungal diversity (one-way ANOVA: $F = 1.12$, $p = 0.354$; Supplementary Fig. 4A) and community composition (PERMANOVA: $R^2 = 0.04$, $p = 0.947$; Supplementary Fig. 4B). The different concentrations of *Colpoda* 2 altered bacterial community composition (ANOSIM: Control vs *Colpoda* 2($10^2$): $R = 0.99$, $p = 0.002$, Control vs *Colpoda* 2($10^4$): $R = 0.99$, $p = 0.003$, *Colpoda* 2($10^2$) vs *Colpoda* 2($10^4$): $R = 0.96$, $p = 0.004$; Fig. 4C), with bacterial Bray-Curtis distance increasing with the inoculation concentrations in comparison to control (Student's *t* test: $T = -13.95$, $p = 6.70\text{E-}22$; Fig. 4D). Compared with the control treatment, *Colpoda* 2 inoculation treatments increased the ratio of pathogen-suppressive bacteria to pathogen-promotive bacteria (*Colpoda* 2 ($10^2$): increase of 120.75%, *Colpoda* 2 ($10^4$): increase of 400.26%, one-way ANOVA with Tukey's HSD test: $F = 224.73$, $p = 6.59\text{E-}12$; Fig. 4E). We further examined the impact of *Colpoda* 2 inoculation on pathogen-suppressive and pathogen-promotive bacterial taxa. The relative abundances of 13 bacterial OTUs were negatively and fifteen positively correlated with the density of *R. solanacearum* (Spearman's correlation analysis). Of these pathogen-suppressive and pathogen-promotive bacterial OTUs, *Colpoda* 2 ($10^2$) and *Colpoda* 2 ($10^4$) increased the relative abundances of seven bacterial OTUs, including *Gp7*, *Marmoricola*, *Gp16*, *Pseudomonas*, *Streptomyces* and *Lysobacter* (one-way ANOVA with Tukey's HSD test: OTU24: $F = 29.59$, $p = 6.00\text{E-}06$; OTU211: $F = 35.92$, $p = 2.00\text{E-}06$; OTU88: $F = 41.33$, $p = 7.90\text{E-}07$; OTU78: $F = 37.65$, $p = 1.00\text{E-}06$; OTU11: $F = 24.48$, $p = 1.90\text{E-}05$; OTU41: $F = 34.60$, $p = 2.00\text{E-}06$; OTU38: $F = 99.34$, $p = 2.23\text{E-}09$; Fig. 4F), and decreased the relative abundances of thirteen bacterial OTUs, including *Gp6*, *Pirellula*, *Planctopirus*, *Chitinophaga*, *Arthrobacter*, *Terrimicrobium*,

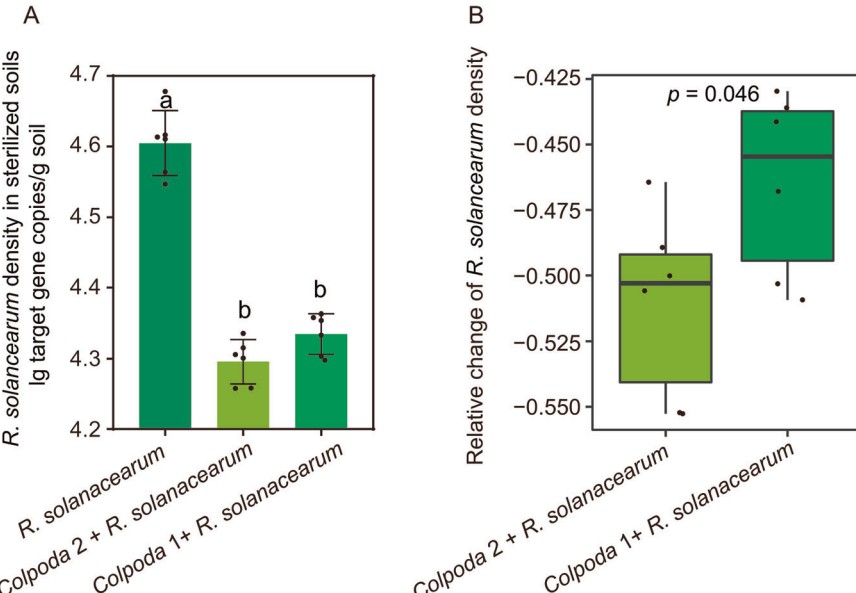

**Fig. 3 | Direct effects of the key predatory protists on pathogen suppression.** Effects of different *Colpoda* on the density of *R. solanacearum* in sterilized soils (**A**). The relative changes of the density of *R. solanacearum* in co-inoculation of different *Colpoda* and *R. solanacearum* combinations (**B**). In (**A**), ANOVA with two-sided Tukey's multiple comparison was used for the statistical analysis. Letters: significant differences between treatments ($p < 0.05$). Results are means ± standard deviation ($n = 6$ biologically independent samples). In (**B**), relative change = (X-control)/control, X = the copies of *R. solanacearum* in co-inoculation of different

*Colpoda* and *R. solanacearum* combinations, control = the copies of *R. solanacearum* in inoculation of *R. solanacearum*. Horizontal bars within boxes represent the median. The tops and bottoms of boxes represent 75th and 25th quartiles, respectively. The upper and lower whiskers represent the range of non-outlier data values. Outliers were plotted as individual points. Significant differences between different treatments are defined by two-sided Student's *t* test ($p < 0.05$). $n = 6$ biologically independent samples. Source data are provided as a Source Data file.

*Kitasatospora*, *Sideroxydans*, *Blastocatella*, *Pseudoduganella* and *Pseudolabrys* (one-way ANOVA with Tukey's HSD test: OTU64: $F = 218.59$, $p = 8.05E\text{-}12$; OTU76: $F = 59.95$, $p = 7.01E\text{-}08$; OTU56: $F = 108.87$, $p = 1.17E\text{-}09$; OTU146: $F = 34.02$, $p = 3.67E\text{-}07$; OTU7: $F = 48.07$, $p = 3.00E\text{-}07$; OTU126: $F = 124.42$, $p = 4.58E\text{-}10$; OTU140: $F = 49.81$, $p = 2.38E\text{-}07$; OTU121: $F = 143.23$, $p = 1.68E\text{-}10$; OTU73: $F = 33.75$, $p = 3.00E\text{-}06$; OTU29: $F = 43.34$, $p = 5.84E\text{-}07$; OTU84: $F = 131.22$, $p = 3.14E\text{-}10$; OTU8: $F = 154.29$, $p = 9.90E\text{-}11$; OTU66: $F = 46.60$, $p = 3.00E\text{-}06$; Fig. 4F).

Among these bacterial OTUs altered by *Colpoda* 2 inoculation (the first greenhouse experiment), the same links of *Pseudomonas*, *Lysobacter*, *Streptomyces*, *Arthrobacter* and *Chitinophaga* with *Colpoda* and *R. solanacearum* were observed in the above field experiment (see above Fig. 2D). Therefore, we selected these key bacterial strains (five *Pseudomonas*, one *Lysobacter*, two *Streptomyces*, two *Arthrobacter* and four *Chitinophaga*) and performed the second greenhouse experiment in sterilized soils to validate the potential links of these bacteria with *Colpoda* 2 and *R. solanacearum*. The relative decrease of bacterial density in *Pseudomonas* 1 + *Colpoda* 2 (decrease of 7.91%), *Pseudomonas* 2 + *Colpoda* 2 (decrease of 9.50%), *Pseudomonas* 3 + *Colpoda* 2 (decrease of 9.57%), *Pseudomonas* 4 + *Colpoda* 2 (decrease of 8.37%), *Pseudomonas* 5 + *Colpoda* 2 (decrease of 9.21%), *Lysobacter* 1 + *Colpoda* 2 (decrease of 10.59%), *Streptomyces* 1 + *Colpoda* 2 (decrease of 9.07%) and *Streptomyces* 2 + *Colpoda* 2 (decrease of 7.04%) were lower than that in *Arthrobacter* 1 + *Colpoda* 2 (decrease of 41.56%), *Arthrobacter* 2 + *Colpoda* 2 (decrease of 38.04%), *Chitinophaga* 1 + *Colpoda* 2 (decrease of 60.35%), *Chitinophaga* 2 + *Colpoda* 2 (decrease of 63.92%), *Chitinophaga* 3 + *Colpoda* 2 (decrease of 61.89%) and *Chitinophaga* 4 + *Colpoda* 2 (decrease of 61.14%) (one-way ANOVA with Tukey's HSD test: $F = 105.21$, $p = 1.54E\text{-}40$; Fig. 5A). In the second part of the second greenhouse experiment, compared with the treatment with inoculation of *R. solanacearum* only, *Pseudomonas* 1 + *R. solanacearum* (decrease of 97.30%), *Pseudomonas* 2 + *R. solanacearum* (decrease of 96.92%), *Pseudomonas* 3 + *R. solanacearum* (decrease of 97.04%),

*Pseudomonas* 4 + *R. solanacearum* (decrease of 97.47%), *Pseudomonas* 5 + *R. solanacearum* (decrease of 96.69%), *Lysobacter* 1 + *R. solanacearum* (decrease of 95.50%), *Streptomyces* 1 + *R. solanacearum* (decrease of 96.92%) and *Streptomyces* 2 + *R. solanacearum* (decrease of 97.71%) decreased the density of *R. solanacearum* and *Arthrobacter* 1 + *R. solanacearum* (increase of 93.44%), *Arthrobacter* 2 + *R. solanacearum* (increase of 139.18%), *Chitinophaga* 1 + *R. solanacearum* (increase of 58.11%), *Chitinophaga* 2 + *R. solanacearum* (increase of 52.90%), *Chitinophaga* 3 + *R. solanacearum* (increase of 73.89%) and *Chitinophaga* 4 + *R. solanacearum* (increase of 57.16%) increased the density of *R. solanacearum* (one-way ANOVA with Tukey's HSD test: $F = 548.13$, $p = 1.82E\text{-}69$; Fig. 5B). The relative changes of densities of bacteria in different bacteria + *Colpoda* 2 treatments were negatively correlated with the relative changes of densities of *R. solanacearum* in different bacteria + *R. solanacearum* treatments (Spearman's correlation analysis: correlation coefficient = −0.73: $p = 1.99E\text{-}15$; Supplementary Table 5).

## Discussion

In this study, we evaluated the impacts of different fertilization regimes on disease suppression and rhizosphere microbiomes within a continuous tomato monoculture cropping system. We provide novel evidence that predatory protists stimulate plant health in response to bioorganic fertilization via direct and indirect reductions of pathogens.

We confirmed hypothesis 1 that protistan communities are more strongly affected by bioorganic fertilization than bacterial and fungal communities. This finding supports previous observations that protistan communities are more sensitive to organic fertilizer inputs than other microbial groups (e.g., bacteria and fungi) in diverse agricultural soils[21,22], potentially explained by the highly diverse taxonomic, trait and functional diversity of protists that together surpasses that of other microbial groups[15,24]. In addition, numerous previous studies demonstrated that long-term different organic fertilization changes

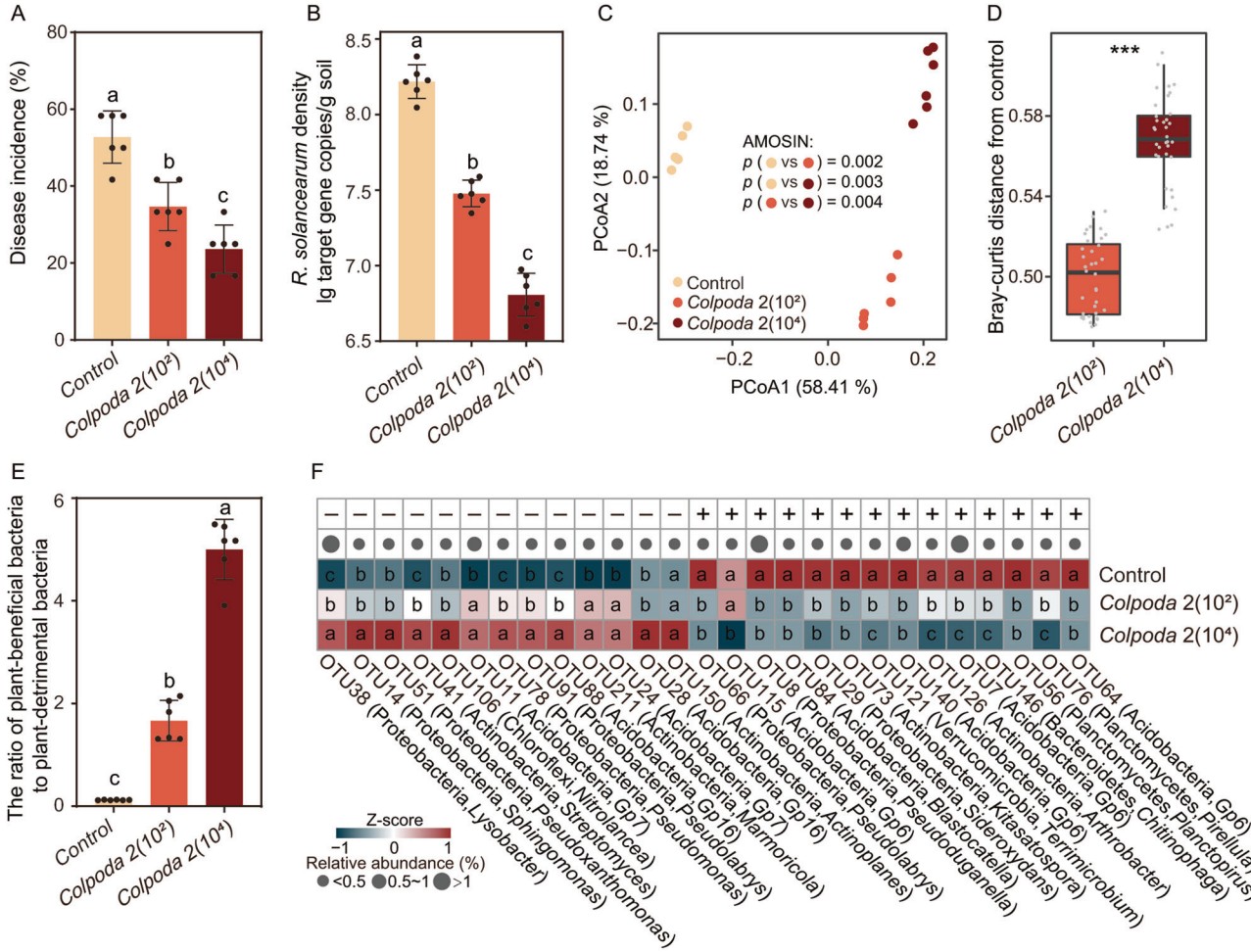

**Fig. 4 | Indirect effects of the key predatory protists on pathogen suppression.** Effects of different concentrations of *Colpoda* 2 on disease incidence (**A**), density of *R. solanacearum* (**B**) and bacterial community composition (**C**). Bray–Curtis distances of bacterial communities between the *Colpoda* 2 inoculation treatments and the control treatment (**D**). Effects of different concentrations of *Colpoda* 2 on the ratio of pathogen-suppressive bacteria to pathogen-promotive bacteria (**E**). Effects of different concentrations of *Colpoda* 2 on the relative abundances of bacterial OTUs (linked with the density of *R. solanacearum*) (**F**). Control = no microbe was inoculated, *Colpoda* 2 ($10^2$) = *Colpoda* 2 was inoculated ($1.0 \times 10^2$ cells $g^{-1}$ dry soil), *Colpoda* 2 ($10^4$) = *Colpoda* 2 was inoculated ($1.0 \times 10^4$ cells $g^{-1}$ dry soil). In (**A**, **B** and **E**), ANOVA with two-sided Tukey's multiple comparison was used for the statistical analysis. Letters: significant differences between treatments ($p < 0.05$). Results are means ± standard deviation ($n = 6$ biologically independent samples). In panel C, significance of bacterial community dissimilarities among different treatments are based on ANOSIM (analysis of similarities, $p < 0.05$). In (**D**), asterisks indicate significant differences as defined by two-sided Student's t test ($p < 0.001$). Horizontal bars within boxes represent the median. The tops and bottoms of boxes represent 75th and 25th quartiles, respectively. The upper and lower whiskers represent the range of non-outlier data values. Outliers were plotted as individual points. $n = 36$ biologically independent samples. In (**F**), the color key relates the heatmap colors to the standard score (z-score). Two-sided Spearman's correlation was performed to explore the relationship between the relative abundances of bacterial OTUs and the density of *R. solanacearum*. Plus signs denote significant positive Spearman's correlations between the relative abundances of bacterial OTUs and the density of *R. solanacearum* ($p < 0.05$). Minus signs denote significant negative Spearman's correlations between the relative abundances of bacterial OTUs and the density of *R. solanacearum* ($p < 0.05$). Circles are proportional to the average relative abundances of bacterial OTUs across all samples. ANOVA with two-sided Tukey's multiple comparison was used for the statistical analysis. Letters: significant differences between treatments ($p < 0.05$). Source data are provided as a Source Data file.

the soil physicochemical environment[25,26]. A stronger response of protistan communities to bioorganic fertilization than of bacterial and fungal communities suggests that protists are more strongly changed, potentially explained by their larger taxonomic, trait and functional diversity might make them more responsive to changes in the surrounding physicochemical environment as induced by bioorganic fertilizer inputs[27,28].

We further confirmed hypothesis 2 that predatory protists explain the decrease in disease incidences and pathogen densities associated with bioorganic fertilization better than other microbial groups. The disease-suppressive function of predatory protists is related to direct and indirect interactions with the bacterial soil-borne pathogen *R. solanacearum*. Previous studies showed that the predominant effects of predatory protists in suppressing soil-borne diseases are

determined by potential negative links between predatory protists and soil-borne pathogenic *R. solanacearum* already at plant establishment[29]. As predatory protists are the dominant, widely distributed soil protistan functional group[19], our results enforce the suppression potential of the community of predatory protists present in soils. In addition, our results complement previous findings that predatory protists could increase plant health effectively through reducing fungal soil-borne pathogens (e.g., *Fusarium oxysporum* and *Rhizoctonia solani*) in other plant-pathogen systems[22,30]. This suggests that predatory protists have important roles in the suppression of soil-borne diseases caused by different pathogens.

We actually identified at least some key predatory protistan taxa underlying disease suppression by directly suppressing bacterial soil-borne pathogens through consumption and thereby confirmed

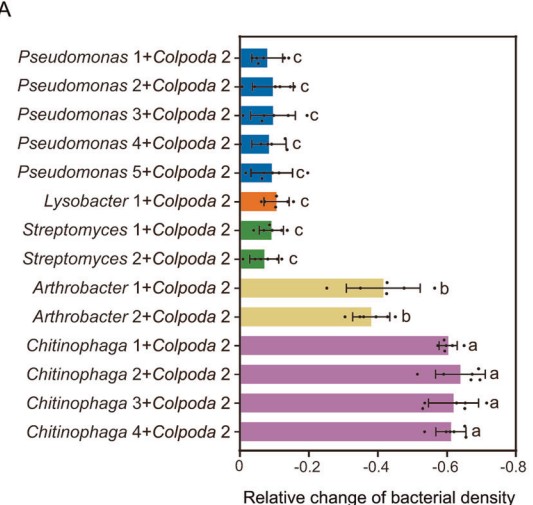

**Fig. 5 | Potential links of key bacteria with *Colpoda* 2 and *R. solanacearum*.** Relative changes of densities of representative bacteria in different key bacteria + *Colpada 2* treatments in the greenhouse experiment using sterilized soils (**A**). Density of *R. solanacearum* in treatments with inoculation of *R. solanacearum* and co-inoculation of representative bacteria in the greenhouse experiment using sterilized soils (**B**). In (**A**), ANOVA with two-sided Tukey's multiple comparison was used for the statistical analysis. Letters: significant differences between treatments (*p* < 0.05). Relative change = (X-control)/control, X = the copies of representative bacteria in different representative bacteria + *Colpoda* 2 treatments, control = the copies of representative bacteria in different representative bacteria inoculation treatments. Representative bacteria are those that are affected by different concentrations of *Colpoda* 2 inoculation and associated with *R. solanacearum* density in field and greenhouse experiments (see above results). Results are means ± standard deviation (*n* = 6 biologically independent samples). In (**B**), ANOVA with two-sided Tukey's multiple comparison was used for the statistical analysis. Letters: significant differences between treatments (*p* < 0.05). Relative change = (X-control)/control, X = the copies of *R. solanacearum* in different bacteria + *R. solanacearum* treatments, control = the copies of *R. solanacearum* in *R. solanacearum* inoculation treatment. Results are means ± standard deviation (*n* = 6 biologically independent samples). Source data are provided as a Source Data file.

hypothesis 3. Some ciliophoran taxa, particularly *Colpoda* spp., fed on *R. solanacearum*, which leads to a direct reduction in pathogen abundance. *Colpoda* is an abundant ciliate genus in soils[27,31] and considered a keystone taxon in soils linked with plant growth (e.g., maize)[27,32,33]. *Colpoda* ciliates feed primarily on bacteria[31]. Previous studies demonstrated that predation of *Colpoda* ciliates can decrease the abundance of some free-living bacteria populations (e.g., *Azospirillum lipoferum* and *Stenotrophomonas* sp.), leading to changes in soil functions (e.g., nitrogen fixation)[34]. Thus, increases in *Colpoda* populations induced by bioorganic fertilization could also reduce pathogenic bacteria, which would enhance plant health. This suggests that predatory protists may be used as effective biocontrol agents to increase soil-borne diseases suppression and *Colpoda* may serve as a keystone genus for future biological control of diseases in diverse agricultural systems.

In addition, we confirmed hypothesis 4 that predatory protists indirectly improve plant health by promoting pathogen-suppressive microorganisms. We show that *Colpoda* ciliates indirectly suppressed *R. solanacearum* through increasing pathogen-suppressive bacteria (e.g., *Pseudomonas*, *Lysobacter* and *Streptomyces*), which produce antimicrobial compounds[35,36]. Simultaneously, *Colpoda* ciliates reduced pathogen-promotive bacteria (e.g., *Arthrobacter* and *Chitinophaga*) that can improve pathogen growth[37]. Species-specific feeding differences between pathogen-suppressive and pathogen-promotive bacteria are likely caused by highly sophisticated antagonism mechanisms of microorganisms against their predators[38] and ancient co-evolutionary predator-prey relationships[16]. Selective predation by protists can decrease pathogen-promotive bacteria, leading to increased niche-space for growth of other microorganisms with traits conferring resistance to protists[18,39]. Among these are those traits that help to defend directly against protist predators and indirectly suppress plant pathogens such as through the production of general antimicrobial compounds[17,22].

We demonstrate that bioorganic fertilization improves plant health by steering the rhizosphere protistan community, particularly

stimulating predatory protists that can directly consume pathogens and indirectly suppressed pathogens by altering bacterial communities in favor of pathogen-suppressive microorganisms. Our study provides mechanistic insights on predatory protists as major agents in controlling plant health that might help in creating more sustainable agricultural practices with considering protists.

## Methods
### Site description, experimental design, tomato bacterial wilt disease incidence determination and rhizosphere soil collection in the field experiment
The site description, experimental design, tomato bacterial wilt disease incidence determination and rhizosphere samples collection is described in detail in[40]. In brief, the field experimental site is located in Hengxi town in Nanjing, Jiangsu Province, China (32°02′N,118°50′E) and the experiment was running since 2013. Three fertilization treatments were established as follows: CF (conventional fertilization), OF (organic fertilization) and BF (bioorganic fertilization). The disease incidence was calculated by counting the number of tomato plants with bacterial wilt (observations of typical wilt symptoms, including necrosis and leaf drooping) among the total number of tomato plants in each plot. The detailed fertilization scheme is shown in Supplementary Table 6. In brief, the scheme of CF (conventional fertilization) is 120 kg ha⁻¹ nitrogen (N), 180 kg ha⁻¹ phosphorus (P) and 120 kg ha⁻¹ potassium (K) mineral fertilizers, the scheme of OF (organic fertilization) is 7500 kg ha⁻¹ organic fertilizer (1.75% nitrogen (N), 0.82% phosphorus (P) and 1.42% potassium (K)) and the scheme of BF (bioorganic fertilization) is 7500 kg ha⁻¹ bioorganic fertilizer (1.85% nitrogen (N), 0.80% phosphorus (P) and 1.46% potassium (K)). Bioorganic fertilizer was produced by inoculating 5% (v/dw) *Bacillus amyloliquefaciens* T-5 into the mixed compost of rapeseed meal and chicken manure (dw/dw = 1:4) compost and then fermented for 1 week. Organic fertilizer was produced using the same process as bioorganic fertilizer, but without inoculation with *Bacillus*. During harvest in June 2018, rhizosphere soil samples were collected and stored at −80 °C for

further use. In brief, plant roots were collected and vigorously shaken in the laboratory. After that, roots were put into a triangular flask with sterile phosphate buffered saline and rhizosphere soil samples were obtained after centrifugation at 10,000 $g$ for 10 min. Then, rhizosphere soil samples were frozen in liquid nitrogen and stored at −80 °C.

## DNA extraction and quantitative PCR assay
The total genomic DNA extraction of rhizosphere soil samples is described in detail in our previous research[40]. In brief, we used PowerSoil Soil DNA Isolation Kits (MoBio Laboratories Inc.,USA) to extract the total genomic DNA of rhizosphere soil samples and the process of the DNA extraction follows the manufacturer's protocol. The abundances of $R. solanacearum$[41], $Pseudomonas$[42], $Lysobacter$[43], $Streptomyces$[44], $Arthrobacter$[45] and $Chitinophaga$[46] were quantified using specific primers sets ($R. solanacearum$: forward, 5′-GAACGCCAA CGGTGCGAACT-3′; reverse, 5′-GGCGGCCTTCAGGGAGGTC-3′; $Pseudomonas$: forward, 5′-GAGTTTGATCCTGGCTCAG-3′; reverse, 5′-GAGT TTGATCCTGGCTCAG-3′; $Lysobacter$: forward, 5′- GAGCCGACGTCGGA TTAGCTGTT-3′; reverse, 5′-AAGGAGGTGWTCCARCC-3′; $Streptomyces$: forward, 5′-GAACTGAGACCGGCTTTTTGA-3′; reverse, 5′-GGTGGCGA AGGCGGA-3′; $Arthrobacter$: forward, 5′-GCTGGTTTGAGAGGACG ACCAGC-3′; reverse, 5′-AGCCCATGACGTTTCTTTCCTGCCA-3′; $Chitinophaga$: forward, 5′-TTRAAGATGGSYGTGCRYC-3′; reverse, 5′-CGC TACATGACATATTCCGCT-3′) and standard curves were generated according to previously established protocols (Supplementary Table 7). Quantitative PCR assay was performed on a qTOWER Real-time PCR system of Analytik Jena (Jena, Germany). Each sample was analyzed in six replicates and the abundance of each microorganism of each sample was expressed as log10 values (target copy numbers per gram soil).

## Illumina MiSeq sequencing and bioinformatic analyses
The prokaryotic 16S rRNA gene amplification and bacterial sequencing library construction is described in ref. [40]. In addition, fungal ITS1 regions and the V4 region of the eukaryotic 18S rRNA gene was amplified with the primer sets ITS1F (5′-CTTGGTCATTTAGAGGA AGTAA-3′)/ITS2 (5′-GCTGCGTTCTTCATCGATGC-3′)[47,48] and V4_1f (5′-C CAGCASCYGCGGTAATWCC-3′)/TAReukREV3 (5′-ACTTTCGTTCTTG ATYYRA-3′)[49], respectively. The fungal and eukaryotic sequencing libraries were constructed according to previously established protocols[29,50]. Paired-end sequencing of all samples was performed on an Illumina MiSeq PE 250 platform at Personal Biotechnology Co., Ltd (Shanghai, China). The raw data of the field and greenhouse experiments are available in the NCBI Sequence Read Archive (SRA) under the BioProject PRJNA957983.

The bacterial, fungal and eukaryotic raw sequences were processed and bacterial, fungal and protistan OTU tables were built with the UPARSE pipeline according to previously established protocols[29,40,50]. In brief, for bacteria and fungi, raw sequences were quality-filtered (sequences with a maximum expected error higher than 0.5 or a length shorter than 200 bp were deleted) and singletons were removed using USEARCH[51]. After that, the remaining sequences were assigned to OTUs at 97% similarity threshold, and chimeras were removed using UCHIME[52]. Finally, the representative sequence of each OTU was matched against the RDP bacterial 16S rRNA gene database and the UNITE fungal ITS database using the RDP classifier[53,54] to obtain the bacterial and fungal OTU tables, respectively. For protists, eukaryotic raw sequences were quality-filtered (sequences with a maximum expected error higher than 0.5 or a length shorter than 350 bp were deleted) and singletons were removed using USEARCH[51]. After that, the remaining sequences were assigned to OTUs at 97% similarity threshold, and chimeras were removed using UCHIME[52]. Finally, the representative sequence of each OTU was matched against the Protist Ribosomal Reference database (PR2)[55]. After that, we removed OTUs assigned as Rhodophyta, Streptophyta, Metazoa, Fungi, unclassified

Opisthokonta and unknown taxa sequences to obtain the protistan OTU table. Based on protistan feeding mode[56,57], protistan OTUs were assigned into different functional groups, including predators, parasites, saprotrophs, plant pathogens, and phototrophs.

## Isolation and identification of rhizosphere predatory protists and bacteria
Because of the observed links between $Colpoda$, bacterial key taxa and disease-suppression in our analyses (see Results), we isolated and identified $Colpoda$ from the rhizosphere soil from the field experiment according to previously described protocols[58] and selected key bacteria isolated in our previous study[40]. For $Colpoda$ strains, the supernatant of homogenized rhizosphere soils was cultivated in each well of 96-well plates (Thermo Fisher, Massachusetts, USA) containing sterile Page's ameba saline[59] (120 mg NaCl, 4 mg $MgSO_4 \cdot 7H_2O$, 4 mg $CaCl_2 \cdot 2H_2O$, 142 mg $Na_2HPO_4$, and 136 mg $KH_2PO_4$ in 1 l of distilled water) buffer (containing inactivated $Escherichia coli$ DH5α as food source of predatory protists) in the dark at 15 °C for 14 days. We performed serial dilutions for the wells to obtain single protist in 96-well plates (Thermo Fisher, Massachusetts, USA), and then selected wells containing pure protistan strains after evaluation with an inverted microscope Nikon Eclipse Ts2 (objective: Nikon CFI Achromat LWD ADL 40X, Ph1, eyepiece: Nikon TS2-W 10X) (NIKON, Tokyo, Japan). The DNA of all pure protistan strains were extracted using the TIANamp Genomic DNA Kit (TIANGEN BIOTECH, Beijing, China) following the manufacturer's instructions. The 18S rRNA gene of protistan strains were amplified using the primers set RibA (5′-ACCTGGTT GATCCTGCCAGT-3′)/RibB (5′-TGATCCATCTGCAGGTTCACCTAC-3′) according to previously established protocols[58]. PCR products were sent for Sanger sequencing in Tsing Ke Biotechnology Co., Ltd. (Wuhan, China). The 18S rRNA gene sequences of pure protistan strains were obtained and blasted against the NCBI GenBank database to obtain their taxonomic information.

## Greenhouse experiments description
The first greenhouse experiment was performed to detect the effects of $Colpoda$ strains on $R. solanacearum$ suppression in the rhizosphere and contained two parts. The first part was performed to detect direct effects of $Colpoda$ strains on $R. solanacearum$ suppression using sterilized soil. The second part was performed with different inoculation amounts of $Colpoda$ 2 to validate concentration effects of this predatory protist on suppressing $R. solanacearum$ by altering microbial communities in nonsterilized soils. The second greenhouse experiment was performed to validate the potential interactions of key bacterial taxa (found from the second part of the first greenhouse experiment and the field experiment) with $Colpoda$ and $R. solanacearum$ using sterilized soils. Detailed processes and inoculation treatments of the first and second greenhouse experiments are shown as follows.

The first part of the first greenhouse experiment was performed in sterilized soils to detect direct effects of $Colpoda$ strains on $Ralstonia solanacearum$ suppression. Treatments in the first part of the first greenhouse experiment as follows: (1) $R. solanacearum$, $Ralstonia solanacearum$ QLRs-1115 ($1.0 \times 10^4$ cells $g^{-1}$ dry soil) was inoculated, (2) $Colpoda$ 2 + $R. solanacearum$, $Colpoda$ strain 2 ($1.0 \times 10^2$ cells $g^{-1}$ dry soil) and $Ralstonia solanacearum$ QLRs-1115 ($1.0 \times 10^4$ cells $g^{-1}$ dry soil) were inoculated, 3) $Colpoda$ 1 + $R. solanacearum$, $Colpoda$ strain 1 ($1.0 \times 10^2$ cells $g^{-1}$ dry soil) and $Ralstonia solanacearum$ QLRs-1115 ($1.0 \times 10^4$ cells $g^{-1}$ dry soil) were inoculated. $Ralstonia solanacearum$ was inoculated two days after seedling transplantation. For co-inoculation treatments of different $Colpoda$ strains and $Ralstonia solanacearum$ combinations, $Ralstonia solanacearum$ was inoculated 2 days after seedling transplanting and $Colpoda$ strains were inoculated 2 weeks after inoculation of $Ralstonia solanacearum$. Rhizosphere samples were collected 3 weeks after the inoculation of protists.

The second part of the first greenhouse experiment was performed with different inoculation concentrations of *Colpoda* 2 in nonsterilized soils to validate indirect effects of this predatory protist on pathogen suppression by altering microbial communities. Treatments in the second part of the first greenhouse experiment as follows: (1) Control, no microbe was inoculated, (2) *Colpoda* 2 (10²), *Colpoda* strain 2 ($1.0 \times 10^2$ cells g⁻¹ dry soil) was inoculated, (3) *Colpoda* 2 (10⁴), *Colpoda* strain 2 ($1.0 \times 10^4$ cells g⁻¹ dry soil) was inoculated. Rhizosphere samples were collected four weeks after the inoculation of protists.

The first part of the second greenhouse experiment was performed in sterilized soils to validate the potential interactions of key bacterial taxa (found from the field experiment and the second part of the first greenhouse experiment) with *Colpoda*. Treatments in the first part of the second greenhouse experiment as follows: (1) *Pseudomonas* 1: *Pseudomonas* strain 1 ($1.0 \times 10^4$ cells g⁻¹ dry soil) was inoculated, (2) *Pseudomonas* 1+ *Colpoda* 2: *Pseudomonas* strain 1 ($1.0 \times 10^4$ cells g⁻¹ dry soil) and *Colpoda* strain 2 ($1.0 \times 10^2$ cells g⁻¹ dry soil) were inoculated, (3) *Pseudomonas* 2: *Pseudomonas* strain 2 ($1.0 \times 10^4$ cells g⁻¹ dry soil) was inoculated, (4) *Pseudomonas* 2+ *Colpoda* 2: *Pseudomonas* strain 2 ($1.0 \times 10^4$ cells g⁻¹ dry soil) and *Colpoda* strain 2 ($1.0 \times 10^2$ cells g⁻¹ dry soil) were inoculated, (5) *Pseudomonas* 3: *Pseudomonas* strain 3 ($1.0 \times 10^4$ cells g⁻¹ dry soil) was inoculated, 6) *Pseudomonas* 3+ *Colpoda* 2: *Pseudomonas* strain 3 ($1.0 \times 10^4$ cells g⁻¹ dry soil) and *Colpoda* strain 2 ($1.0 \times 10^2$ cells g⁻¹ dry soil) were inoculated, (7) *Pseudomonas* 4: *Pseudomonas* strain 4 ($1.0 \times 10^4$ cells g⁻¹ dry soil) was inoculated, (8) *Pseudomonas* 4+ *Colpoda* 2: *Pseudomonas* strain 4 ($1.0 \times 10^4$ cells g⁻¹ dry soil) and *Colpoda* strain 2 ($1.0 \times 10^2$ cells g⁻¹ dry soil) were inoculated, (9) *Pseudomonas* 5: *Pseudomonas* strain 5 ($1.0 \times 10^4$ cells g⁻¹ dry soil) was inoculated, 10) *Pseudomonas* 5+ *Colpoda* 2: *Pseudomonas* strain 5 ($1.0 \times 10^4$ cells g⁻¹ dry soil) and *Colpoda* strain 2 ($1.0 \times 10^2$ cells g⁻¹ dry soil) were inoculated, 11) *Lysobacter* 1: *Lysobacter* strain 1 ($1.0 \times 10^4$ cells g⁻¹ dry soil) was inoculated, 12) *Lysobacter* 1+ *Colpoda* 2: *Lysobacter* strain 1 ($1.0 \times 10^4$ cells g⁻¹ dry soil) and *Colpoda* strain 2 ($1.0 \times 10^2$ cells g⁻¹ dry soil) were inoculated, 13) *Streptomyces* 1: *Streptomyces* strain 1 ($1.0 \times 10^4$ cells g⁻¹ dry soil) was inoculated, 14) *Streptomyces* 1+ *Colpoda* 2: *Streptomyces* strain 1 ($1.0 \times 10^4$ cells g⁻¹ dry soil) and *Colpoda* strain 2 ($1.0 \times 10^2$ cells g⁻¹ dry soil) were inoculated, 15) *Streptomyces* 2: *Streptomyces* strain 2 ($1.0 \times 10^4$ cells g⁻¹ dry soil) was inoculated, 16) *Streptomyces* 2+ *Colpoda* 2: *Streptomyces* strain 2 ($1.0 \times 10^4$ cells g⁻¹ dry soil) and *Colpoda* strain 2 ($1.0 \times 10^2$ cells g⁻¹ dry soil) were inoculated, 17) *Arthrobacter* 1: *Arthrobacter* strain 1 ($1.0 \times 10^4$ cells g⁻¹ dry soil) was inoculated, 18) *Arthrobacter* 1+ *Colpoda* 2: *Arthrobacter* strain 1 ($1.0 \times 10^4$ cells g⁻¹ dry soil) and *Colpoda* strain 2 ($1.0 \times 10^2$ cells g⁻¹ dry soil) were inoculated, 19) *Arthrobacter* 2: *Arthrobacter* strain 2 ($1.0 \times 10^4$ cells g⁻¹ dry soil) was inoculated, 20) *Arthrobacter* 2+ *Colpoda* 2: *Arthrobacter* strain 2 ($1.0 \times 10^4$ cells g⁻¹ dry soil) and *Colpoda* strain 2 ($1.0 \times 10^2$ cells g⁻¹ dry soil) were inoculated, 21) *Chitinophaga* 1: *Chitinophaga* strain 1 ($1.0 \times 10^4$ cells g⁻¹ dry soil) was inoculated, 22) *Chitinophaga* 1+ *Colpoda* 2: *Chitinophaga* strain 1 ($1.0 \times 10^4$ cells g⁻¹ dry soil) and *Colpoda* strain 2 ($1.0 \times 10^2$ cells g⁻¹ dry soil) were inoculated, 23) *Chitinophaga* 2: *Chitinophaga* strain 2 ($1.0 \times 10^4$ cells g⁻¹ dry soil) was inoculated, 24) *Chitinophaga* 2+ *Colpoda* 2: *Chitinophaga* strain 2 ($1.0 \times 10^4$ cells g⁻¹ dry soil) and *Colpoda* strain 2 ($1.0 \times 10^2$ cells g⁻¹ dry soil) were inoculated, 25) *Chitinophaga* 3: *Chitinophaga* strain 3 ($1.0 \times 10^4$ cells g⁻¹ dry soil) was inoculated, 26) *Chitinophaga* 3+ *Colpoda* 2: *Chitinophaga* strain 3 ($1.0 \times 10^4$ cells g⁻¹ dry soil) and *Colpoda* strain 2 ($1.0 \times 10^2$ cells g⁻¹ dry soil) were inoculated, 27) *Chitinophaga* 4: *Chitinophaga* strain 4 ($1.0 \times 10^4$ cells g⁻¹ dry soil) was inoculated, 28) *Chitinophaga* 4+ *Colpoda* 2: *Chitinophaga* strain 4 ($1.0 \times 10^4$ cells g⁻¹ dry soil) and *Colpoda* strain 2 ($1.0 \times 10^2$ cells g⁻¹ dry soil) in were inoculated. Different bacteria were inoculated two days after seedling transplanting. For co-inoculation treatments of different *Colpoda* strains and bacteria combinations, different bacteria were inoculated two days after seedling transplanting and *Colpoda* strains were inoculated

two weeks after inoculation of bacteria. Rhizosphere samples were collected three weeks after the inoculation of protists.

The the second part of second greenhouse experiment was performed in sterilized soils to validate the potential interactions of key bacterial taxa (found from the field experiment and the second part of the first greenhouse experiment) with *Ralstonia solanacearum*. Treatments in the second part of the second greenhouse experiment as follows: 1) *R. solanacearum*: *Ralstonia solanacearum* QLRs-1115 ($1.0 \times 10^5$ cells g⁻¹ dry soil) was inoculated, 2) *Pseudomonas* 1 + *R. solanacearum*: *Pseudomonas* strain 1 ($1.0 \times 10^5$ cells g⁻¹ dry soil) and *Ralstonia solanacearum* QLRs-1115 ($1.0 \times 10^5$ cells g⁻¹ dry soil) were inoculated, 3) *Pseudomonas* 2 + *R.solanacearum*: *Pseudomonas* strain 2 ($1.0 \times 10^5$ cells g⁻¹ dry soil) and *Ralstonia solanacearum* QLRs-1115 ($1.0 \times 10^5$ cells g⁻¹ dry soil) was inoculated, 4) *Pseudomonas* 3 + *R. solanacearum*: *Pseudomonas* strain 3 ($1.0 \times 10^5$ cells g⁻¹ dry soil) and *Ralstonia solanacearum* QLRs-1115 ($1.0 \times 10^5$ cells g⁻¹ dry soil) were inoculated, 5) *Pseudomonas* 4 + *R. solanacearum*: *Pseudomonas* strain 4 ($1.0 \times 10^5$ cells g⁻¹ dry soil) and *Ralstonia solanacearum* QLRs-1115 ($1.0 \times 10^5$ cells g⁻¹ dry soil) were inoculated, 6) *Pseudomonas* 5 + *R. solanacearum*: *Pseudomonas* strain 5 ($1.0 \times 10^5$ cells g⁻¹ dry soil) and *Ralstonia solanacearum* QLRs-1115 ($1.0 \times 10^5$ cells g⁻¹ dry soil) were inoculated, 7) *Lysobacter* 1 + *R. solanacearum*: *Lysobacter* strain 1 ($1.0 \times 10^5$ cells g⁻¹ dry soil) and *Ralstonia solanacearum* QLRs-1115 ($1.0 \times 10^5$ cells g⁻¹ dry soil) were inoculated, 8) *Streptomyces* 1 + *R. solanacearum*: *Streptomyces* strain 1 ($1.0 \times 10^5$ cells g⁻¹ dry soil) and *Ralstonia solanacearum* QLRs-1115 ($1.0 \times 10^5$ cells g⁻¹ dry soil) were inoculated, 9) *Streptomyces* 2 + *R. solanacearum*: *Streptomyces* strain 2 ($1.0 \times 10^5$ cells g⁻¹ dry soil) and *Ralstonia solanacearum* QLRs-1115 ($1.0 \times 10^5$ cells g⁻¹ dry soil) were inoculated, 10) *Arthrobacter* 1 + *R. solanacearum*: *Arthrobacter* strain 1 ($1.0 \times 10^5$ cells g⁻¹ dry soil) and *Ralstonia solanacearum* QLRs-1115 ($1.0 \times 10^5$ cells g⁻¹ dry soil) were inoculated, 11) *Arthrobacter* 2 + *R. solanacearum*: *Arthrobacter* strain 2 ($1.0 \times 10^5$ cells g⁻¹ dry soil) and *Ralstonia solanacearum* QLRs-1115 ($1.0 \times 10^5$ cells g⁻¹ dry soil) were inoculated, 12) *Chitinophaga* 1 + *R. solanacearum*: *Chitinophaga* strain 1 ($1.0 \times 10^5$ cells g⁻¹ dry soil) and *Ralstonia solanacearum* QLRs-1115 ($1.0 \times 10^5$ cells g⁻¹ dry soil) were inoculated, 13) *Chitinophaga* 2 + *R. solanacearum*: *Chitinophaga* strain 2 ($1.0 \times 10^5$ cells g⁻¹ dry soil) and *Ralstonia solanacearum* QLRs-1115 ($1.0 \times 10^5$ cells g⁻¹ dry soil) were inoculated, 14) *Chitinophaga* 3 + *R. solanacearum*: *Chitinophaga* strain 3 ($1.0 \times 10^5$ cells g⁻¹ dry soil) and *Ralstonia solanacearum* QLRs-1115 ($1.0 \times 10^5$ cells g⁻¹ dry soil) were inoculated, 15) *Chitinophaga* 4 + *R. solanacearum*: *Chitinophaga* strain 4 ($1.0 \times 10^5$ cells g⁻¹ dry soil) and *Ralstonia solanacearum* QLRs-1115 ($1.0 \times 10^5$ cells g⁻¹ dry soil) were inoculated. Two days after seedling transplanting, different bacteria were inoculated and *Ralstonia solanacearum* was inoculated 2 weeks after inoculation of different bacteria. Rhizosphere samples were collected 3 weeks after the inoculation of *Ralstonia solanacearum*.

Soils of the greenhouse experiments were collected from the fields treated with conventional fertilization. All soils were taken back to the laboratory and air-dried in the shade. After that, soils were passed through a 2-mm sieve to ensure soil homogenization in subsequent greenhouse experiments. Soils were sterilized by Co75 γ-ray irradiation (65 kGy) at Xiyue Technology Co., Ltd (Nanjing, China). Each of the six replicates contained 12 sterilized tomato seedlings (cultivar "Hong ai sheng") and each seedling was planted in a black polypropylene pot containing 300 g dry soil. All pots were cultivated in a greenhouse (daytime: 16 h and average 30 °C, night: 8 h and average 25 °C) at the Nanjing Agriculture University. Pots were watered with deionized water every 2 days and with ½ strength Hoagland nutrient solution (Hopebio, Qingdao, China) every 10 days. The pots were weekly randomized throughout the experiments. Rhizosphere soil samples were collected at the end of each greenhouse experiment according to the method as described above. DNA extractions, quantitative PCR assays and Illumina MiSeq sequencing of rhizosphere soil

samples, bioinformatic analyses of data and disease incidence determination were according to methods described above.

## Statistical analyses

We used nonparametric Shannon indexes (calculated by MOTHUR)[60,61] to estimate bacterial, fungal and protistan α-diversities. We used principal coordinate analysis (PCoA) based on the Bray–Curtis distance to compare the differences of bacterial, fungal and protistan community compositions at the OTU level in R (R version 4.0.1). We used permutational multivariate analysis of variance (PERMANOVA)[62] to assess the effects of different fertilization regimes on the community compositions of rhizosphere microorganisms by the adonis function with 999 permutations using R package "vegan"[63] (R version 4.0.1). We used one-way ANOVA to assess effects of different fertilization regimes on the α-diversities of rhizosphere microorganisms using SPSS v20.0 (SPSS Inc. USA). ANOSIM (analysis of similarities, calculated by MOTHUR)[61,64] was performed to investigate significant differences of bacterial community composition between different treatments in the greenhouse experiments. We selected bacterial, fungal and protistan α-diversities (Shannon index) and community compositions (PCoA1) as main microbial predictors to calculate the significance of effects of these predictors on disease incidence using multiple regression by linear models in R (R version 4.0.1). The explanatory power and significance of each microbial predictor was assessed using R package "relaimpo"[65] (R version 4.0.1). We used Spearman's correlation coefficient to evaluate correlations between abundant predatory protistan OTUs (average relative abundance >0.1%) and disease incidence, key predatory protistan OTUs (linked with disease incidence) and abundant bacterial OTUs (average relative abundance >0.1%) in the field experiment, and abundant bacterial OTUs (average relative abundance >0.1%) and the density of *R. solanacearum* in the greenhouse experiments, respectively. The correlation coefficients and *p* values of Spearman's correlations were calculated through the "corr.test" function using R package "psych"[66] (R version 4.0.1). We used heatmaps to show Spearman's correlations between key predatory protistan OTUs (linked with disease incidence) and abundant bacterial OTUs (average relative abundance >0.1%) using R package "pheatmap"[67] (R version 4.0.1). Mantel test was conducted to evaluate the correlations between rhizosphere bacterial and fungal communities and the relative abundances of rhizosphere predatory protists using R package "vegan"[63] (R version 4.0.1). One-way ANOVA with Tukey's HSD test was used for multiple comparisons and student's *t*-test was performed to compare significances of the difference between distinct treatments in SPSS v22.0 (SPSS Inc. USA). For pathogen-suppressive microorganisms and pathogen-promotive microorganisms, putative pathogen-suppressive microorganisms were defined as those with relative abundances that negatively correlated with the density of *R. solanacearum* (Spearman's correlation: $p < 0.05$), and putative pathogen-promotive microorganisms were defined as those with relative abundances that positively correlated with the density of *R. solanacearum* (Spearman's correlation: $p < 0.05$). The formula for the ratio of pathogen-suppressive microorganisms to pathogen-promotive microorganisms was SUM-PPS/SUM-PPP, where SUM-PPS is the sum of the relative abundances of putative pathogen-suppressive microorganisms and SUM-PPP is the sum of the relative abundances of putative pathogen-promotive microorganisms.

## Reporting summary

Further information on research design is available in the Nature Portfolio Reporting Summary linked to this article.

## Data availability

The raw data used in this study are available in the NCBI Sequence Read Archive (SRA) under accession code BioProject PRJNA957983. Source data are provided with this paper.

## Code availability

All codes used in this study are available on GitHub (https://github.com/SaiGuo92/Code-for-paper1) and Zendo[68].

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

## Acknowledgements
This work was supported by the Fundamental Research Funds for the Central Universities (KYT2023001 (to R.L.), QTPY2023003 (to R.L.), XUEKEN2023031 (to H.L.) and XUEKEN2023039 (to Q.S.)), the National Natural Science Foundation of China (42277294 (to R.L.), 42307171 (to S.G.) and 42377294 (to X.D.)), the Postdoctoral Science Foundation of China (2023M731724) (to S.G.), the Excellent Postdoctoral Program of Jiangsu Province, China (2023ZB706) (to S.G.), the Guidance Foundation, the Hainan Institute of Nanjing Agricultural University (NAUSY-MS10) (to R.L.),the Achievement Transformation Fund project of Hainan Research Institute of Nanjing Agricultural University (NAUSY-CG-ZD-01) (to R.L.) and the Natural Science Foundation of Jiangsu Province, China (BK20210390) (to C.T.).

## Author contributions
S.G., R.L., Q.S., G.A.K. and S.G. developed the ideas and designed the experimental plans. S.G., Z.J., Z.Y., X.Y., X.D., C.T. and H.L. performed the experiments. S.G., W.X., R.L. and S.G. analyzed the data. S.G., R.L., Q.S., G.A.K. and S.G. participated in the completion of the manuscript.

## Competing interests
The authors declare no competing interests.
