## [Peer Review File · Nature Communications]

Reviewers' Comments:

Reviewer #1:

Remarks to the Author:

The manuscript by Guo et al. describes novel results demonstrating that predatory protists enhance plant health via direct and indirect reductions of the major tomato bacterial wilt pathogen *Ralstonia solanacearum*, an important soilborne pathogen of solanaceous crops. The authors combined amplicon sequencing, qPCR and plate culturing, and integrated the data well. The work describes a lot of observations, experiments and discoveries.

The results are interesting and potentially important in a context where the importance of predatory protists to plant health has been only marginally reported. Firstly, the authors used a long-term field experiment to find reduced disease incidences and pathogen densities were mainly determined by increased predatory protists, particularly *Colpoda* sp., and their potential links with indigenous bacteria. Subsequently, the authors isolated predatory protists and bacteria, and performed a series of greenhouse experiments to validate observed results in the field experiment and explore the potential disease-suppressive mechanisms of predatory protists (predatory protists directly consumed pathogens and indirectly suppressed pathogens by altering bacterial communities in favor of pathogen-suppressive microorganisms).

In general, this manuscript is compelling, logical and rich in content. Though this study is very novel and attractive, the manuscript has some points to improve. Thus, I encourage the authors to make the necessary effort to improve along the suggested lines below.

Simplified the present hypothesizes, now it is not concise enough, more like a statement of the results. hypothesis 1 needs to be refined, and the term 'fertilization practices' is too large because the differences in fertilization treatments in the study are only reflected in organic fertilizers, and a more relevant assumption should be made (see references <https://doi.org/10.1186/s40168-019-0647-0>, in which it was found that applying nitrogen fertilizer reduces soil microbial diversity and has a more significant impact on bacteria and fungi than primary predatory protozoa.).

Introduction :

Line 43 I suggest removing "can"

Line 46 I suggest changing "reduce" to "reduced"

Line 55 I suggest changing "soil" to "soils"

Line 60 "... predatory protists as catalyzers of nutrient turnover" What nutrient? Nitrogen? The description here is somewhat general. Please precise it.

Line 70 I think the word "disease-suppressive mechanisms" needs to be precise. Your study only investigates bacterial wilt disease. So, I suggest changing "disease-suppressive mechanisms" to "bacterial wilt disease-suppressive mechanisms".

Results :

Line 78 For the title "Microbial communities and their contribution to disease incidence" Microbial communities not only include bacteria, fungi and protists, but also include viruses and other microorganisms. I think this title needs to be corrected. Please precise it.

Line 81-83 I think the description of this part is a little complicated. For the sentence "bacterial diversity: one-way ANOVA: $F = 0.51$, $p > 0.05$, fungal diversity: one-way ANOVA: $F = 0.99$, $p > 0.05$; bacterial community composition: PERMANOVA: $R^2 = 0.37$, $p > 0.05$, fungal community composition: PERMANOVA: $R^2 = 0.26$, $p > 0.05$ ", I suggest deleting F and R^2 , just show the p-value.

Line 92 I find that you show positive correlation between predatory protists and bacterial community using Mantel test. Mantel correlation cannot show positive or negative. I suggest removing "positively" here.

Line 94 Abbreviations that appear for the first time in the manuscript should require a clear description. This makes it clearer to the reader. For example, change "BF" to "BF (bioorganic fertilization)". "CF" and "OF" should be modified as well.

Line 113 I suggest removing "was"

Line 127 I suggest changing "...different *Colpoda* and *R. solanacearum* combinations..." to "different *Colpoda* strains and *R. solanacearum* combinations..." Your greenhouse experiments use isolated *Colpoda* strains, if you just say "*Colpoda*", the reader will be confused with *Colpoda* OTU.

Line 148 In the Methods, you use the term "Bray-Curtis". Please keep the word consistent

throughout the text. For the term "bray-curtis", "b" and "c" need to be capitalized, here.

Discussion:

Line 199 "other microbial groups" What microbial groups? Bacteria? Fungi? Precise this point is better.

Line 200 A higher diverse microbial community is considered to have higher stability, why is it more sensitive to fertilization inputs? Increase relevant discussions to explain that protozoa have stronger responsiveness to fertilization treatment.

Line 202 "A" The font should be Times New Roman.

Line 208 I don't understand the word "different" in this sentence. I suggest removing "different" or explaining it.

Line 218 In the Results, you show that Ciliophoran taxa are key protists that affect plant health. So, I suggest adding the word "key" here, like "some key predatory protistan taxa"

Line 222 You discuss the key predatory protists, Colpoda, and cite references to show that Colpoda is linked with plant growth. What kind of plant? Precise this point is better for the clarity of the Discussion.

Line 230-233 You show that Colpoda ciliates indirectly suppressed *R. solanacearum* through increasing pathogen-suppressive bacteria and reduced pathogen-promotive bacteria. For this result, you discuss a lot about the predation relationship between predatory protists and pathogen-suppressive and pathogen-promotive bacteria (e.g. species-specific feeding differences and selective predation). Indeed, it is necessary to discuss this point. I think microbial competition in ecological niches is also a key point to discuss. The earth is inhabited by a vast quantity of diverse microorganisms. Microorganisms may engage in ferocious competition, resulting in a relentless war to win over finite resources such as nutrients, light or territory. There may be niche competition between pathogen-promotive bacteria and pathogen-suppressive bacteria. Because of species-specific feeding differences and selective predation, predatory protists prey on pathogen-promotive bacteria easily than on pathogen-suppressive bacteria. Predatory protists prey on pathogen-promotive bacteria (that is, prey on niche competitors of pathogen-suppressive bacteria), releasing more niche for pathogen-suppressive bacteria to grow in soils. I think you can add 1-2 sentences to discuss this point.

Methods:

The sampling was not detailed enough and did not explain whether the health status of the plant samples in each treatment was consistent. Because even though the disease severity in each fertilization treatment is different, there will be healthy plants and diseased plants in each treatment, so the health status of plants during sampling will also have a certain impact on the sample results.

Line 253 Even though the methods were described previously, the authors still need to give important information in the Methods. A short summary containing information is essential for the readers. Although the authors wrote a short summary, tomato bacterial wilt disease incidence determination and detailed fertilization scheme are missing in this short summary.

Please add it.

Line 294 I suggest changing "link" to "links".

Line 331 For the Methods, I think you should specify what software you use, and this will make it clearer to the readers. Which software was used to calculate the α -diversity? Please precise it.

Line 338 Same problem as above. Which software was used to calculate ANOSIM (analysis of similarities)? Please precise it.

Reviewer #2:

Remarks to the Author:

This manuscript entitled " Predatory protists impact plant health via direct and indirect reductions of *Ralstonia solanacearum* " focused on the dominant roles in microbiome (bacteria, fungi and protists), especially for the protists, linking soil-borne disease suppression, and demonstrates nicely the key role of protists (especially predatory protists) in pathogen suppression in the whole microbiome. The authors document changes in the microbiome communities (bacteria, fungi and protists) in different long-term fertilization treatments using a metabarcoding approach, as well as

link the microbiome communities with disease incidence and the pathogen *R. solanacearum*. The authors further definite the key disease-suppressive microbial group (predatory protists) and lock the key disease-suppressive predatory protistan taxa. Then, a number of analyses to seek the direct and indirect suppressive effects of predatory protists on the plant pathogen *R. solanacearum*. After that, the authors isolate the predatory protists and bacteria from the rhizosphere and adopt greenhouse experiments testing the direct and indirect suppressive effects of predatory protists on pathogens. At last, the authors clarify microbial mechanisms underlying fertilization-induced increases in plant health from a holistic microbiome view and concluded that predatory protists are the key disease-suppressive microbial group and can directly consume pathogens and indirectly suppressed pathogens by altering bacterial communities in favor of pathogen-suppressive microorganisms. Although there are many reports on the links between bacteria and/or fungi and disease suppression, the effects of other microbial component-protist and interactions between protists and other microorganisms is largely elusive. Observations in this manuscript address an important knowledge gap in our understanding of the role of the entire microbiome and its interactions in determining tomato bacterial wilt. It significantly moves the research field forward. The manuscript is well written, novel and convincing. Before publication, I suggest the authors fix some minor details. My comments as below:

Lines 24-25: Authors said "investigated rhizosphere bacteria, fungi and predatory protists over eleven growing seasons". However, the authors investigate the communities of bacteria, fungi and protist in the rhizosphere, so please revise it. "predatory protists"-> "protists"

Line 27: The word " relative abundances of" can be removed, as this could make the Abstract more concise.

Line 41: "Pathogens lead to a serious impact on plant health...", authors can give some examples, like "Pathogens (such as...) lead to a serious impact on plant health..."

Line 47: The sentence can be revised to this: "Healthy plants are essential for ensuring crop productivity and food security." The words " maintaining " and " ensuring " are somewhat repetitive in this sentence.

Line 58: "...even more than their microbial prey...", authors can give some examples, like "...even more than their microbial prey (such as ...)..." This manuscript is novel, so try to introduce more clear for the readers.

Line 78: "disease incidence" can be changed to "bacterial wilt disease incidence" in this subtitle. It is more precise to the readers. After that, you can use the word "disease incidence". The readers can understand more easily.

Line 97: Same as before. "disease suppression" can be changed to "bacterial wilt disease suppression"

Lines 195-196: Here, the authors used the word "fertilization practices", but the author kept using the word "fertilization regimes" in the previous text. Please be consistent.

Line 202: "protist" should be changed to "protistan"

Line 210: and 211 "soilborne" should be changed to "soil-borne"

Line 220: Add the word "direct" to the sentence as "a direct reduction". This could separate from the indirect effect that follows.

Line 298: Please specify the Page's Amoeba Saline (pas), and add a reference.

Line 300: please add detailed information for 96-well plates. Which country? Company?

Line 325: The pots were periodically randomized? Please mention.

Reviewer #3:

Remarks to the Author:

This study investigated the potential role of predatory protists in improving plant health, based on multiple growing seasons of tomato under various fertilization regimes with different bacterial wilt disease incidences. They found the relationships between reduction in disease incidences and pathogen densities under bioorganic fertilization were best predicted by increase in the abundances of predatory protists. They also constructed microcosm experiments to explore the mechanisms underlying the links of predatory protists with plant health. They concluded that predatory protists can directly ingest pathogens and indirectly suppress pathogens through modulating the plant beneficial bacterial communities.

This study provides novel insights into how predatory protists can be used as potential agrifood tools to improve plant health and support sustainable agriculture. The manuscript is well written and easy to follow. I have the following comments and suggestions for authors to further improve their manuscript.

Line 50-54 It would be great to clarify whether these studies have also reported the potential role of protists in improving plant health

Line 56-57 Predatory protists can be also directly affected by bacteria and fungi, which are their main food resources

Line 61-62 I disagree that the importance of predatory protists for plant health are little studied. You have cited a number of publications on this topic. Better reword this sentence and highlight the novelty of the current study

Line 70-72 Any potential mechanisms why you hypothesized that protists are more responsive to fertilization than bacteria and fungi?

Line 70-75 You have too many hypotheses to test in the current study

Line 79 You may need to add a brief introduction of the experiment in one sentence before you start introducing the results

Line 110 Please clarify how you defined and identified "pathogen suppressive bacteria" and "pathogen promotive bacteria"?

Line 127-130 It is important to clarify how you inoculated these combinations? These strains need to adapt to a new environment, and thus it is important whether they (i.e. *R. solanacearum*) have stabilized in the soil environment before you inoculate the protist stains

Line 124 Have you tested the suppression effects of predatory protists on pathogens in medium in the laboratory?

Line 137 The presence of pathogens is not directly related to disease outbreak. Have you considered to explore the direct effects of protists in reducing the disease incidences?

Line 195-205 I don't think you need to discuss this phenomenon, which has been reported before and is not new. Focus on the main findings of your current study

Line 258-259 Do you have results from two sampling years/seasons. It would make the results more robust and convincing, particularly from a long-term field experiment

Line 319 It is not clear what is "conventional fields"?

Line 324 "300 g soil" is a very low amount of soil for glasshouse experiment, especially for the tomato plants

Responses to reviewers' comments

REVIEWER COMMENTS

Reviewer #1 (Remarks to the Author):

The manuscript by Guo et al. describes novel results demonstrating that predatory protists enhance plant health via direct and indirect reductions of the major tomato bacterial wilt pathogen *Ralstonia solanacearum*, an important soilborne pathogen of solanaceous crops. The authors combined amplicon sequencing, qPCR and plate culturing, and integrated the data well. The work describes a lot of observations, experiments and discoveries.

The results are interesting and potentially important in a context where the importance of predatory protists to plant health has been only marginally reported. Firstly, the authors used a long-term field experiment to find reduced disease incidences and pathogen densities were mainly determined by increased predatory protists, particularly *Colpoda* sp., and their potential links with indigenous bacteria. Subsequently, the authors isolated predatory protists and bacteria, and performed a series of greenhouse experiments to validate observed results in the field experiment and explore the potential disease-suppressive mechanisms of predatory protists (predatory protists directly consumed pathogens and indirectly suppressed pathogens by altering bacterial communities in favor of pathogen-suppressive microorganisms).

In general, this manuscript is compelling, logical and rich in content. Though this study is very novel and attractive, the manuscript has some points to improve. Thus, I encourage the authors to make the necessary effort to improve along the suggested lines below.

Response: Thank you very much for your positive evaluation. We revised our manuscript according to your constructive and helpful comments as detailed below.

Simplified the present hypothesizes, now it is not concise enough, more like a statement of the results. hypothesis 1 needs to be refined, and the term 'fertilization practices' is too large because the differences in fertilization treatments in the study are only reflected in organic fertilizers, and a more relevant assumption should be made (see references <https://doi.org/10.1186/s40168-019-0647-0>, in which it was found that applying nitrogen fertilizer reduces soil microbial diversity and has a more significant impact on bacteria and fungi than primary predatory protozoa.).

Response: We agree and revised our previous hypotheses. These now read as follows: “We hypothesized that (1) protistan communities are more strongly affected by bioorganic fertilization than bacterial and fungal communities, (2) predatory protists explain the decrease in disease incidences and pathogen densities associated with bioorganic fertilization better than other microbial groups, (3) predatory protists directly suppress *R. solanacearum* through consumption and (4) indirectly improve plant health by promoting pathogen-suppressive microorganisms.”

Introduction:

Line 43 I suggest removing “can”

Response: We deleted “can”.

Line 46 I suggest changing “reduce” to “reduced”

Response: We replaced “reduce” by “reduced” as suggested.

Line 55 I suggest changing “soil” to “soils”

Response: We replaced “soil” by “soils” as suggested.

Line 60 “... predatory protists as catalyzers of nutrient turnover” What nutrient? Nitrogen? The description here is somewhat general. Please precise it.

Response: We revised this sentence – it now reads as follows: “Predatory protists stimulate nutrient turnover (e.g. nitrogen) resulting in increased plant-availability¹⁶”

References:

16. Gao, Z., Karlsson, I., Geisen, S., Kowalchuk, G. & Jousset, A. Protists: puppet masters of the rhizosphere microbiome. *Trends Plant Sci.* 24, 165–176 (2019).

Line 70 I think the word “disease-suppressive mechanisms” needs to be precise. Your study only investigates bacterial wilt disease. So, I suggest changing “disease-suppressive mechanisms” to “bacterial wilt disease-suppressive mechanisms”.

Response: We replaced “disease-suppressive mechanisms” by “bacterial wilt disease-suppressive mechanisms”.

Results:

Line 78 For the title “Microbial communities and their contribution to disease incidence” Microbial communities not only include bacteria, fungi and protists, but also include viruses and other microorganisms. I think this title needs to be corrected. Please precise it.

Response: We revised this subtitle – it now reads as follows: “Bacterial, fungal and protistan communities and their contribution to bacterial wilt disease incidence.”

Line 81-83 I think the description of this part is a little complicated. For the sentence “bacterial diversity: one-way ANOVA: $F = 0.51$, $p > 0.05$, fungal diversity: one-way ANOVA: $F = 0.99$, $p > 0.05$; bacterial community composition: PERMANOVA: $R^2 = 0.37$, $p > 0.05$, fungal community composition: PERMANOVA: $R^2 = 0.26$, $p > 0.05$ ”, I suggest deleting F and R^2 , just show the p -value.

Response: In the guide sent by the editor of Nature Communications the policy states that the name of the statistical test, the value of the test statistic, and the p -value or R^2 should be provided along with any statistical value given. So, we retained the F -value and R^2 .

Line 92 I find that you show positive correlation between predatory protists and bacterial community using Mantel test. Mantel correlation cannot show positive or negative. I suggest removing “positively” here.

Response: We deleted “positively”.

Line 94 Abbreviations that appear for the first time in the manuscript should require a clear description. This makes it clearer to the reader. For example, change “BF” to “BF (bioorganic fertilization)”. “CF” and “OF” should be modified as well.

Response: We replaced “BF” by “BF (bioorganic fertilization)”, “CF” by “CF (conventional fertilization)” and “OF” by “OF (organic fertilization)”.

Line 113 I suggest removing “was”

Response: We deleted “was”.

Line 127 I suggest changing “...different Colpoda and *R. solanacearum* combinations...” to “different Colpoda strains and *R. solanacearum* combinations...” Your greenhouse experiments use isolated Colpoda strains, if you just say “Colpoda”, the reader will be confused with Colpoda OTU.

Response: We replaced “Colpoda” by “Colpoda strains”.

Line 148 In the Methods, you use the term “Bray-Curtis”. Please keep the word consistent throughout the text. For the term “bray-curtis”, “b” and “c” need to be capitalized, here.

Response: We revised this word as suggested – it now reads as follows: “bacterial Bray-Curtis distance increasing with the inoculation concentrations in comparison to control.”

Discussion:

Line 199 “other microbial groups” What microbial groups? Bacteria? Fungi? Precise this point is better.

Response: We revised this sentence – it now reads as follows: “This finding supports previous observations that protistan communities are more sensitive to organic fertilizer inputs than other microbial groups (e.g., bacteria and fungi) in diverse agricultural soils^{21,22}”

References:

21. Guo, S. et al. Protists as main indicators and determinants of plant performance. *Microbiome* 9, 64 (2021).

22. Guo, S. et al. Trophic interactions between predatory protists and pathogen-suppressive bacteria impact plant health.

ISME J. 1–12 (2022).

Line 200 A higher diverse microbial community is considered to have higher stability, why is it more sensitive to fertilization inputs? Increase relevant discussions to explain that protozoa have stronger responsiveness to fertilization treatment.

Response: Previous studies demonstrate that protists are more diverse in traits like size than bacteria with sizes ranging in many orders of magnitude (Geisen et al., 2017). Additionally, protists are phylogenetically diverse and contain virtually all eukaryotes with the exception of plants, fungi and metazoan (Geisen et al., 2018). Functionally, protists also contain many nutrient uptake modes ranging from predators to phototrophs to parasites (Geisen et al., 2018). This makes the response of distinct protist taxa to changes in the surrounding very unique. These are summarized in the Discussion section. It now reads as follows: “This finding supports previous observations that protistan communities are more sensitive to organic fertilizer inputs than other microbial groups (e.g., bacteria and fungi) in diverse agricultural soils ^{21,22}, potentially explained by the highly diverse taxonomic, trait and functional diversity of protists that together surpasses that of other microbial groups ^{15,24}”

References:

15. Geisen, S. et al. Soil protists: a fertile frontier in soil biology research. *FEMS Microbiol. Rev.* 42, 293–323 (2018).
21. Guo, S. et al. Protists as main indicators and determinants of plant performance. *Microbiome* 9, 64 (2021).
22. Guo, S. et al. Trophic interactions between predatory protists and pathogen-suppressive bacteria impact plant health. *ISME J.* 1–12 (2022).
24. Geisen, S. et al. Soil protistology rebooted: 30 fundamental questions to start with. *Soil Biol. Biochem.* 111, 94–103 (2017).

Line 202 “A” The font should be Times New Roman.

Response: We revised this word as suggested.

Line 208 I don't understand the word “different” in this sentence. I suggest removing “different” or explaining it.

Response: We deleted “different”.

Line 218 In the Results, you show that Ciliophoran taxa are key protists that affect plant health. So, I suggest adding the word “key” here, like “some key predatory protistan taxa”

Response: We added “key”.

Line 222 You discuss the key predatory protists, Colpoda, and cite references to show that Colpoda is linked with plant growth. What kind of plant? Precise this point is better for the clarity of the Discussion.

Response: We revised this sentence – it now reads as follows: “*Colpoda* is an abundant ciliate genus in soils ^{27,31} and considered a keystone taxon in soils linked with plant growth (e.g., maize) ^{27,32,33}”

References:

27. Foissner, W. Protozoa as bioindicators in agroecosystems, with emphasis on farming practices, biocides, and biodiversity. *Agric. Ecosyst. Environ.* 62, 93–103 (1997).
31. Foissner, W. An updated compilation of world soil ciliates (Protozoa, Ciliophora), with ecological notes, new records, and descriptions of new species. *Eur. J. Protistol.* 34, 195–235 (1998).
32. Bamforth, S. S. Protozoa: recyclers and indicators of agroecosystem quality. in *Fauna in Soil Ecosystems* (CRC Press, 1997).
33. Zhang, W., Lin, Q., Li, G. & Zhao, X. The ciliate protozoan *Colpoda cucullus* can improve maize growth by transporting soil phosphates. *J. Integr. Agric.* 21, 855–861 (2022).

Line 230-233 You show that Colpoda ciliates indirectly suppressed *R. solanacearum* through increasing pathogen-suppressive bacteria and reduced pathogen-promotive bacteria. For this result, you discuss a lot about the predation relationship between predatory protists and pathogen-suppressive and pathogen-promotive bacteria (e.g. species-specific feeding differences and selective predation). Indeed, it is necessary to discuss this point. I think microbial competition in ecological niches is also a key point to discuss. The earth is inhabited by a vast quantity of diverse microorganisms. Microorganisms may engage in

ferocious competition, resulting in a relentless war to win over finite resources such as nutrients, light or territory. There may be niche competition between pathogen-promotive bacteria and pathogen-suppressive bacteria. Because of species-specific feeding differences and selective predation, predatory protists prey on pathogen-promotive bacteria easily than on pathogen-suppressive bacteria. Predatory protists prey on pathogen-promotive bacteria (that is, prey on niche competitors of pathogen-suppressive bacteria), releasing more niche for pathogen-suppressive bacteria to grow in soils. I think you can add 1-2 sentences to discuss this point.

Response: We improved the discussion as suggested. It now reads as follows: “Species-specific feeding differences between pathogen-suppressive and pathogen-promotive bacteria are likely caused by highly sophisticated antagonism mechanisms of microorganisms against their predators³⁸ and ancient co-evolutionary predator-prey relationships¹⁶. Selective predation by protists can decrease pathogen-promotive bacteria, leading to increased niche-space for growth of other microorganisms with traits conferring resistance to protists^{18,39}. Among these are those traits that help to defend directly against protist predators and indirectly suppress plant pathogens such as through the production of general antimicrobial compounds^{17,22}.”

References:

16. Gao, Z., Karlsson, I., Geisen, S., Kowalchuk, G. & Jousset, A. Protists: puppet masters of the rhizosphere microbiome. *Trends Plant Sci.* 24, 165–176 (2019).
17. Jousset, A. & Bonkowski, M. The model predator *Acanthamoeba castellanii* induces the production of 2, 4, DAPG by the biocontrol strain *Pseudomonas fluorescens* Q2-87. *Soil Biol. Biochem.* 42, 1647–1649 (2010).
18. Jousset, A. et al. Predators promote defence of rhizosphere bacterial populations by selective feeding on non-toxic cheaters. *ISME J.* 3, 666–674 (2009).
22. Guo, S. et al. Trophic interactions between predatory protists and pathogen-suppressive bacteria impact plant health. *ISME J.* 1–12 (2022) doi:10.1038/s41396-022-01244-5.
38. Jousset, A. Ecological and evolutive implications of bacterial defences against predators. *Environ. Microbiol.* 14, 1830–1843 (2012).
39. Bauer, M. A., Kainz, K., Carmona-Gutierrez, D. & Madeo, F. Microbial wars: competition in ecological niches and within the microbiome. *Microb. Cell* 5, 215 (2018).

Methods:

The sampling was not detailed enough and did not explain whether the health status of the plant samples in each treatment was consistent. Because even though the disease severity in each fertilization treatment is different, there will be healthy plants and diseased plants in each treatment, so the health status of plants during sampling will also have a certain impact on the sample results.

Response: As rhizosphere samples collection was described in detail in our previous study, we cited this paper to avoid redundant descriptions in the present manuscript. The detailed sampling process was as follows: for the rhizosphere samples, 3 tomato roots were collected from three random points in each plot (see following figure). These tomato roots were vigorously shaken to remove excess soil. The soil adhering to the roots (rhizosphere soil) was collected by sterile phosphate buffered saline (PBS). The rhizosphere soil samples were obtained after centrifugation at 10000 g for 10 min, and stored at –80 °C for soil DNA extraction.

Figure. Overview of the field experiment, sample collection and subsequent sample processing conducted in this study.

Line 253 Even though the methods were described previously, the authors still need to give important information in the Methods. A short summary containing information is essential for the readers. Although the authors wrote a short summary, tomato bacterial wilt disease incidence determination and detailed fertilization scheme are missing in this short summary. Please add it.

Response: We revised this part as suggested. It now reads as follows: “The disease incidence was calculated by counting the number of tomato plants with bacterial wilt (observations of typical wilt symptoms, including necrosis and leaf drooping) among the total number of tomato plants in each plot. The detailed fertilization scheme is shown in Supplementary Table 6. In brief, the scheme of CF (conventional fertilization) is 120 kg ha⁻¹ nitrogen (N), 180 kg ha⁻¹ phosphorus (P) and 120 kg ha⁻¹ potassium (K) mineral fertilizers, the scheme of OF (organic fertilization) is 7500 kg ha⁻¹ organic fertilizer (1.75% nitrogen (N), 0.82% phosphorus (P) and 1.42% potassium (K)) and the scheme of BF (bioorganic fertilization) is 7500 kg ha⁻¹ bioorganic fertilizer (1.85% nitrogen (N), 0.80% phosphorus (P) and 1.46% potassium (K)).”

Line 294 I suggest changing “link” to “links”.

Response: We replaced “link” by “links”.

Line 331 For the Methods, I think you should specify what software you use, and this will make it clearer to the readers. Which software was used to calculate the α -diversity? Please precise it.

Response: We used the software MOTHUR to calculate the α -diversity and added this software name in the text. It now reads as follows: “We used non-parametric Shannon indexes (calculated by MOTHUR)^{60,61} to estimate bacterial, fungal and protistan α -diversities.”

References:

60. Chao, A. & Shen, T.-J. Nonparametric estimation of Shannon’s index of diversity when there are unseen species in sample. *Environ. Ecol. Stat.* 10, 429–443 (2003).
61. Kozich, J. J., Westcott, S. L., Baxter, N. T., Highlander, S. K. & Schloss, P. D. Development of a dual-index sequencing strategy and curation pipeline for analyzing amplicon sequence data on the MiSeq Illumina sequencing platform. *Appl. Environ. Microbiol.* 79, 5112–5120 (2013).

Line 338 Same problem as above. Which software was used to calculate ANOSIM (analysis of similarities)? Please precise it.

Response: Same as above. It now reads as follows: “ANOSIM (analysis of similarities, calculated by MOTHUR)^{61,64} was performed to investigate significant differences of bacterial community composition between different treatments in the greenhouse experiments.”

References:

61. Kozich, J. J., Westcott, S. L., Baxter, N. T., Highlander, S. K. & Schloss, P. D. Development of a dual-index sequencing strategy and curation pipeline for analyzing amplicon sequence data on the MiSeq Illumina sequencing platform. *Appl. Environ. Microbiol.* 79, 5112–5120 (2013).
64. Clarke, K. R. Non-parametric multivariate analyses of changes in community structure. *Aust. J. Ecol.* 18, 117–143 (1993).

Reviewer #2 (Remarks to the Author):

This manuscript entitled " Predatory protists impact plant health via direct and indirect reductions of *Ralstonia solanacearum* " focused on the dominant roles in microbiome (bacteria, fungi and protists), especially for the protists, linking soil-borne disease suppression, and demonstrates nicely the key role of protists (especially predatory protists) in pathogen suppression in the whole microbiome. The authors document changes in the microbiome communities (bacteria, fungi and protists) in different long-term fertilization treatments using a metabarcoding approach, as well as link the microbiome communities with disease incidence and the pathogen *R. solanacearum*. The authors further definite the key disease-suppressive microbial group (predatory protists) and lock the key disease-suppressive predatory protistan taxa. Then, a number of analyses to seek the direct and indirect suppressive effects of predatory protists on the plant pathogen *R. solanacearum*. After that, the authors

isolate the predatory protists and bacteria from the rhizosphere and adopt greenhouse experiments testing the direct and indirect suppressive effects of predatory protists on pathogens. At last, the authors clarify microbial mechanisms underlying fertilization-induced increases in plant health from a holistic microbiome view and concluded that predatory protists are the key disease-suppressive microbial group and can directly consume pathogens and indirectly suppressed pathogens by altering bacterial communities in favor of pathogen-suppressive microorganisms. Although there are many reports on the links between bacteria and/or fungi and disease suppression, the effects of other microbial component-protist and interactions between protists and other microorganisms is largely elusive. Observations in this manuscript address an important knowledge gap in our understanding of the role of the entire microbiome and its interactions in determining tomato bacterial wilt. It significantly moves the research field forward. The manuscript is well written, novel and convincing. Before publication, I suggest the authors fix some minor details. My comments as below:

Response: Thank you very much for your positive evaluation. We revised our manuscript according to your constructive and helpful comments as detailed below.

Lines 24-25: Authors said “investigated rhizosphere bacteria, fungi and predatory protists over eleven growing seasons”. However, the authors investigate the communities of bacteria, fungi and protist in the rhizosphere, so please revise it. “predatory protists”-> “protists”

Response: We revised this sentence – it now reads as follows: “To elucidate the role of importance of predatory importance in affecting plant health and the underlying mechanisms, we investigated rhizosphere bacteria, fungi and protists over eleven growing seasons of tomato planting under conventional, organic, and bioorganic fertilization regimes with different bacterial wilt disease incidences.”

Line 27: The word " relative abundances of" can be removed, as this could make the Abstract more concise.

Response: We deleted “relative abundances of”.

Line 41: “Pathogens lead to a serious impact on plant health...”, authors can give some examples, like “Pathogens (such as...) lead to a serious impact on plant health...”

Response: We revised this sentence – it now reads as follows: “Pathogens (e.g. *Ralstonia solanacearum* and *Fusarium oxysporum*) severely impact plant health when colonizing the plant rhizosphere ³.”

References:

3. Mendes, R., Garbeva, P. & Raaijmakers, J. M. The rhizosphere microbiome: significance of plant beneficial, plant pathogenic, and human pathogenic microorganisms. *FEMS Microbiol. Rev.* 37, 634–663 (2013).

Line 47: The sentence can be revised to this: “Healthy plants are essential for ensuring crop productivity and food security.” The words " maintaining " and " ensuring " are somewhat repetitive in this sentence.

Response: We revised this sentence – it now reads as follows: “Healthy plants are essential for ensuring crop productivity and food security ⁹.”

References:

9. Flood, J. The importance of plant health to food security. *Food Secur.* 2, 215–231 (2010).

Line 58: “...even more than their microbial prey...”, authors can give some examples, like “...even more than their microbial prey (such as ...)...” This manuscript is novel, so try to introduce more clear for the readers.

Response: We revised this sentence – it now reads as follows: “Therefore, predatory protist communities are sensitive to fertilizer application, potentially even more than their microbial prey (e.g. bacteria and fungi) ^{21,22}.”

References:

21. Guo, S. et al. Protists as main indicators and determinants of plant performance. *Microbiome* 9, 64 (2021).

22. Guo, S. et al. Trophic interactions between predatory protists and pathogen-suppressive bacteria impact plant health. *ISME J.* 1–12 (2022).

Line 78: “disease incidence” can be changed to “bacterial wilt disease incidence” in this subtitle. It is more precise to the

readers. After that, you can use the word “disease incidence”. The readers can understand more easily.

Response: We replaced “disease incidence” by “bacterial wilt disease incidence”.

Line 97: Same as before. “disease suppression” can be changed to “bacterial wilt disease suppression”

Response: We replaced “disease suppression” by “bacterial wilt disease suppression”.

Lines 195-196: Here, the authors used the word “fertilization practices”, but the author kept using the word “fertilization regimes” in the previous text. Please be consistent.

Response: We revised this sentence – it now reads as follows: “We confirmed hypothesis 1 that protistan communities are more strongly affected by bioorganic fertilization than bacterial and fungal communities.”

Line 202: “protist” should be changed to “protistan”

Response: We replaced “protist” by “protistan”.

Line 210: and 211 “soilborne” should be changed to “soil-borne”

Response: We replaced “soilborne” by “soil-borne”.

Line 220: Add the word “direct” to the sentence as “a direct reduction”. This could separate from the indirect effect that follows.

Response: We revised this sentence – it now reads as follows: “Some ciliophoran taxa, particularly *Colpoda* spp., fed on *R. solanacearum*, which leads to a direct reduction in pathogen abundance.”

Line 298: Please specify the Page's Amoeba Saline (pas), and add a reference.

Response: We revised this part and added a reference – it now reads as follows: “For *Colpoda* strains, the supernatant of homogenized rhizosphere soils was cultivated in each well of 96-well plates (Thermo Fisher, Massachusetts, USA) containing sterile Page’s amoeba saline⁵⁹ (120 mg NaCl, 4 mg MgSO₄·7H₂O, 4 mg CaCl₂·2H₂O, 142 mg Na₂HPO₄, and 136 mg KH₂PO₄ in 1 liter of distilled water) buffer (containing inactivated *Escherichia coli* DH5α as food source of predatory protists) in the dark at 15 °C for 14 days.”

References:

59. Thomas, V., Herrera-Rimann, K., Blanc, D. S. & Greub, G. Biodiversity of amoebae and amoeba-resisting bacteria in a hospital water network. *Appl. Environ. Microbiol.* 72, 2428–2438 (2006).

Line 300: please add detailed information for 96-well plates. Which country? Company?

Response: We added detailed information about 96-well plates in the Methods. It now reads as follows: “We performed serial dilutions for the wells to obtain single protist in 96-well plates (Thermo Fisher, Massachusetts, USA).”

Line 325: The pots were periodically randomized? Please mention.

Response: We added a sentence in the Methods. It now reads as follows: “The pots were weekly randomized throughout the experiments.”

Reviewer #3 (Remarks to the Author):

This study investigated the potential role of predatory protists in improving plant health, based on multiple growing seasons of tomato under various fertilization regimes with different bacterial wilt disease incidences. They found the relationships between reduction in disease incidences and pathogen densities under bioorganic fertilization were best predicted by increase in the abundances of predatory protists. They also constructed microcosm experiments to explore the mechanisms underlying the links of predatory protists with plant health. They concluded that predatory protists can directly ingest pathogens and indirectly suppress pathogens through modulating the plant beneficial bacterial communities.

This study provides novel insights into how predatory protists can be used as potential agrifood tools to improve plant health

and support sustainable agriculture. The manuscript is well written and easy to follow. I have the following comments and suggestions for authors to further improve their manuscript.

Response: Thank you very much for your positive evaluation. We revised our manuscript according to your constructive and helpful comments as detailed below.

Line 50-54 It would be great to clarify whether these studies have also reported the potential role of protists in improving plant health

Response: We clarified and revised this part that now reads as follows: “Organic farming, such as organic fertilizer application, can improve plant performance (e.g. growth and health) and minimize negative impacts of synthetic chemicals by inducing beneficial bacteria and fungi as well as their ecological interactions in soils and plant rhizospheres ^{8,12,13}.”

References:

8. Liu, H. et al. Continuous application of different organic additives can suppress tomato disease by inducing the healthy rhizospheric microbiota through alterations to the bulk soil microflora. *Plant Soil* 423, 229–240 (2018).
12. Maeder, P. et al. Soil fertility and biodiversity in organic farming. *Science* 296, 1694–1697 (2002).
13. Schmidt, J. E., Kent, A. D., Brisson, V. L. & Gaudin, A. C. M. Agricultural management and plant selection interactively affect rhizosphere microbial community structure and nitrogen cycling. *Microbiome* 7, 146 (2019).

Line 56-57 Predatory protists can be also directly affected by bacteria and fungi, which are their main food resources

Response: We have revised this part, which now reads: “Bacteria and fungi are also top-down controlled by predators, particularly predatory protists that are the dominant soil protists ¹⁵. In turn, bacteria and fungi affect predatory protists. For example, anti-predatory compounds released by bacteria and fungi commonly inhibit protists ^{16,17}. Also species-specific interactions between protist predators and microbial prey are common, as predators specifically select for their preferred prey microorganisms ^{15,16,18}.”

References:

15. Geisen, S. et al. Soil protists: a fertile frontier in soil biology research. *FEMS Microbiol. Rev.* 42, 293–323 (2018).
16. Gao, Z., Karlsson, I., Geisen, S., Kowalchuk, G. & Jousset, A. Protists: puppet masters of the rhizosphere microbiome. *Trends Plant Sci.* 24, 165–176 (2019).
17. Jousset, A. & Bonkowski, M. The model predator *Acanthamoeba castellanii* induces the production of 2, 4, DAPG by the biocontrol strain *Pseudomonas fluorescens* Q2-87. *Soil Biol. Biochem.* 42, 1647–1649 (2010).
18. Jousset, A. et al. Predators promote defence of rhizosphere bacterial populations by selective feeding on non-toxic cheaters. *ISME J.* 3, 666–674 (2009).

Line 61-62 I disagree that the importance of predatory protists for plant health are little studied. You have cited a number of publications on this topic. Better reword this sentence and highlight the novelty of the current study

Response: We rewrote this sentence – it now reads as follows: “While an increasing number of studies uncover the importance of protists in the complexity of disease suppression and therefore as contributors to plant health ^{15,16}, the mode of action how these predatory protists increase plant health remains unknown.”

References:

15. Geisen, S. et al. Soil protists: a fertile frontier in soil biology research. *FEMS Microbiol. Rev.* 42, 293–323 (2018).
16. Gao, Z., Karlsson, I., Geisen, S., Kowalchuk, G. & Jousset, A. Protists: puppet masters of the rhizosphere microbiome. *Trends Plant Sci.* 24, 165–176 (2019).

Line 70-72 Any potential mechanisms why you hypothesized that protists are more responsive to fertilization than bacteria and fungi?

Response: Previous studies demonstrate that protists are more diverse in traits like size than bacteria with sizes ranging in many orders of magnitude (Geisen et al., 2017). Additionally, protists are phylogenetically diverse and contain virtually all eukaryotes with the exception of plants, fungi and metazoan (Geisen et al., 2018). Functionally, protists also contain many nutrient uptake modes ranging from predators to phototrophs to parasites (Geisen et al., 2018). This makes the response of distinct protist taxa to changes in the surrounding very unique.

References:

Geisen, S. et al. Soil protists: a fertile frontier in soil biology research. *FEMS Microbiol. Rev.* 42, 293–323 (2018).

Geisen, S. et al. Soil protistology rebooted: 30 fundamental questions to start with. *Soil Biol. Biochem.* 111, 94–103 (2017).

Line 70-75 You have too many hypotheses to test in the current study

Response: Thank you very much for your comments. Considering the complexity of our design and structure of our article, we prefer to keep four hypotheses. The main reason is that we want specific and testable ones rather than broad ones that in several ways can provide support or contradictions to a single hypothesis.

Line 79 You may need to add a brief introduction of the experiment in one sentence before your start introducing the results

Response: This has been included as suggested. The text now reads: “The long-term field experiment was performed with different fertilization treatments (conventional, organic and bioorganic fertilization) that resulted in different disease incidence.”

Line 110 Please clarify how you defined and identified “pathogen suppressive bacteria” and “pathogen promotive bacteria”?

Response: We have now added definitions and ways to identify “pathogen suppressive bacteria” and “pathogen promotive bacteria” in the Methods. The text reads as follows: “Putative pathogen-suppressive microorganisms were defined as those with relative abundances that negatively correlated with the density of *R. solanacearum* (Spearman’s correlation: $p < 0.05$), and putative pathogen-promotive microorganisms were defined as those with relative abundances that positively correlated with the density of *R. solanacearum* (Spearman’s correlation: $p < 0.05$).” In addition, we have isolated representative pathogen-suppressive and pathogen-promotive microbial taxa and validated their pathogen-suppressive and pathogen-promotive abilities (see Fig.5B).

Fig. 5 Relative changes in densities of representative bacteria in different representative bacteria + *Colpoda* 2 treatments in the greenhouse experiment using sterilized soils (A). Density of *R. solanacearum* in treatments with *R. solanacearum* and co-inoculation of representative bacteria in the greenhouse experiment using sterilized soils (B).

In panel A, bars with different letters indicate significant differences between different treatments as defined by one-way ANOVA with Tukey's HSD test ($p < 0.05$). Relative change = $(X - \text{control}) / \text{control}$, X = the copies of representative bacteria in different representative bacteria + *Colpoda* 2 treatments, control = the copies of representative bacteria in different representative bacteria inoculation treatments. Representative bacteria are those that are affected by different concentrations of *Colpoda* 2 inoculation and associated with *R. solanacearum* density in field and greenhouse experiments (see above results). Results are means \pm SD (n = 6). In panel B, bars with different letters indicate significant differences between different treatments as defined by one-way ANOVA with Tukey's HSD test ($p < 0.05$). Relative change = $(X - \text{control}) / \text{control}$, X = the copies of *R. solanacearum* in different bacteria + *R. solanacearum* treatments, control = the copies of *R.*

solanacearum in *R. solanacearum* inoculation treatment. Results are means \pm SD (n = 6).

Line 127-130 It is important to clarify how you inoculated these combinations? These strains need to adapt to a new environment, and thus it is important whether they (i.e. *R. solanacearum*) have stabilized in the soil environment before you inoculate the protist stains

Response: We added the detailed explanation of the inoculation process in the Methods. The text now reads: “*Ralstonia solanacearum* was inoculated two days after seedling transplantation. For co-inoculation treatments of different *Colpoda* strains and *Ralstonia solanacearum* combinations, *Ralstonia solanacearum* was inoculated two days after seedling transplanting and *Colpoda* strains were inoculated two weeks after inoculation of *Ralstonia solanacearum*.” We also set up a control treatment (only *Ralstonia solanacearum* inoculation) in the greenhouse experiment. At the end of the experiment, we collected the rhizosphere samples of control and co-inoculation treatments and measured *Ralstonia solanacearum* density.

Line 124 Have you tested the suppression effects of predatory protists on pathogens in medium in the laboratory?

Response: We have tested the suppression effects of predatory protists on pathogens in medium (Page’s amoeba saline (containing 0.5% Nutrient Broth)) in the laboratory before. At the end of the experiment, a standard 10-fold dilution plating assay was used to count the number of *R. solanacearum* using selective media (SMSA) (Elphinstone et al., 1996). The result showed that different *Colpoda* strains and *R. solanacearum* combinations decreased the number of *R. solanacearum* compared to the control (only *R. solanacearum*), which is in line with our greenhouse experiment (see following figure).

Figure. Effects of the two *Colpoda* strains on colony forming units of *R. solanacearum* in medium (A). The relative changes of the number of *R. solanacearum* in co-inoculation of different *Colpoda* and *R. solanacearum* combinations (B).

In panel A, bars with different letters indicate significant differences between different treatments as defined by one-way ANOVA with Tukey's HSD test ($p < 0.05$). In panel B, relative change = $(X - \text{control}) / \text{control}$, X = the copies of *R. solanacearum* in co-inoculation of different *Colpoda* and *R. solanacearum* combinations, control = the copies of *R. solanacearum* in inoculation of *R. solanacearum*.

References:

Elphinstone J, Hennessy J, Wilson J, et al. Sensitivity of different methods for the detection of *Ralstonia solanacearum* in potato tuber extracts. EPPO Bulletin, 1996, 26: 663-678

Line 137 The presence of pathogens is not directly related to disease outbreak. Have you considered to explore the direct effects of protists in reducing the disease incidences?

Response: We have explored this effect and showed it as follows: “Compared with the control treatment, the different concentrations of *Colpoda 2* decreased disease incidence (*Colpoda 2* (10^2): decrease of 34.21%, *Colpoda 2* (10^4): decrease of 55.26%, one-way ANOVA with Tukey’s HSD test: $F = 31.20$, $p = 5.00E-06$; Fig. 4A).”

Line 195-205 I don’t think you need to discuss this phenomenon, which has been reported before and is not new. Focus on the main findings of your current study

Response: Thank you very much for your comments. We agree you that this point is not novel. Yet, we designed the study to test the related hypotheses as not many studies focused on different responsiveness of large microbial groups. We believe that confirmatory studies, especially when they are a small part of the overall study, are key to test for generalities.

Line 258-259 Do you have results from two sampling years/seasons. It would make the results more robust and convincing, particularly from a long-term field experiment

Response: The soil treated with different fertilizer have shown stable disease suppression in the long-term field experiment. We have chosen the longest year for sampling that we could collect at the time. Thus, in selecting 2018 to decipher the ecological mechanisms of soil organisms to protect plant health, we have chosen a representative year that is reflective of the longer terms trends.

Line 319 It is not clear what is “conventional fields”?

Response: We revised this part, which now reads: “Soils of the greenhouse experiments were collected from the fields treated with conventional fertilization”

Line 324 “300 g soil” is a very low amount of soil for glasshouse experiment, especially for the tomato plants

Response: Previous studies have demonstrated similar use of low amount of soil (200-300g soil per pot or plant) to plant tomato in greenhouse experiments (Hu et al., 2021; Deng et al., 2022; Li et al., 2022; Yang et al., 2023). In particular, we used a kind of commercial little tomato cultivar “Hong ai sheng”, which is a dwarf tomato with short growth cycle. So, a pot containing 300g of dry soil should be suitable for cultivating tomato plants in greenhouse experiments.

References:

Hu J, Yang T, Friman V P, et al. Introduction of probiotic bacterial consortia promotes plant growth via impacts on the resident rhizosphere microbiome. *Proceedings of the Royal Society B*, 2021, 288(1960): 20211396

Deng X, Zhang N, Li Y, et al. Bio-organic soil amendment promotes the suppression of *Ralstonia solanacearum* by inducing changes in the functionality and composition of rhizosphere bacterial communities. *New Phytologist*, 2022, 235(4): 1558-1574.

Li M, Pommier T, Yin Y, et al. Indirect reduction of *Ralstonia solanacearum* via pathogen helper inhibition. *The ISME journal*, 2022, 16(3): 868-875.

Yang K, Fu R, Feng H, et al. RIN enhances plant disease resistance via root exudate-mediated assembly of disease-suppressive rhizosphere microbiota. *Molecular Plant*, 2023.

Reviewers' Comments:

Reviewer #1:

Remarks to the Author:

The authors improved the manuscript. All the comments have been addressed.

Reviewer #2:

Remarks to the Author:

The authors have answered all my questions, and significantly improved the manuscript. I have no further requests. I believe this work will make great contribution to the scientific community.

Reviewer #3:

Remarks to the Author:

All my major comments and suggestions have been addressed. I have no more concerns.

Responses to reviewers' comments

Reviewer #1 (Remarks to the Author):

The authors improved the manuscript. All the comments have been addressed.

Response: Thank you very much.

Reviewer #2 (Remarks to the Author):

The authors have answered all my questions, and significantly improved the manuscript. I have no further requests. I believe this work will make great contribution to the scientific community.

Response: Thank you very much

Reviewer #3 (Remarks to the Author):

All my major comments and suggestions have been addressed. I have no more concerns.

Response: Thank you very much.